# Constraint-guided Hardware-aware NAS through Gradient Modification

**Gregory De Ruyter[1,2], Mathias Verbeke[1,2,3], Hans Hallez[1,2]**
[1]Department of Computer Science, KU Leuven, Belgium
[2]Leuven.AI - KU Leuven Institute for AI, Belgium [3]Flanders Make@KU Leuven
{gregory.deruyter,mathias.verbeke,hans.hallez}@kuleuven.be

## Abstract

Neural Architecture Search (NAS), particularly gradient-based techniques, has proven highly effective in automating the design of neural networks. Recent work has extended NAS to hardware-aware settings, aiming to discover architectures that are both accurate and computationally efficient. Many existing methods integrate hardware metrics into the optimization objective as regularization terms, which introduces differentiability requirements and hyperparameter tuning challenges. This can either result in overly penalizing resource-intensive architectures or architectures failing to meet the hardware constraints of the target device. To address these challenges, we propose CONNAS, a novel gradient-based NAS framework that enforces hardware constraints directly through gradient modification. This approach eliminates the need for differentiable hardware metrics and regularization weights. The novelty in CONNAS lies in modifying gradients with respect to architectural choices, steering the search away from infeasible architectures while ensuring constraint satisfaction. Evaluations on the NATS-Bench benchmark demonstrate that CONNAS consistently discovers architectures that meet the imposed hardware constraints while achieving performance within just $0.14\%$ of the optimal feasible architecture. Additionally, in a practical deployment scenario, CONNAS outperforms handcrafted architectures by up to $1.55\%$ in accuracy under tight hardware budgets. Our code is publicly available at https://gitlab.kuleuven.be/m-group-campus-brugge/distrinet_public/connas.

## 1 Introduction

Deep neural networks have proven to be very successful in numerous applications ranging from image recognition (Krizhevsky et al., 2017) and speech recognition (Hinton et al., 2012) to time series segmentation (Lea et al., 2017). More recently, the use of deep neural networks on constrained hardware has gained significant interest in the context of machine learning on the edge (EdgeML), where resource-constrained devices such as mobile devices, embedded computers, and microcontrollers are used to perform inference tasks. These devices often have limited computational power, memory, and energy resources. As a result, it is crucial to design neural network architectures that operate effectively under strict hardware constraints.

A key challenge in EdgeML is designing neural network architectures that meet hardware constraints while maintaining high performance. In recent years, Neural Architecture Search (NAS) was introduced to automate the design of neural networks, and to potentially discover novel architectures that outperform manually designed models by experts (Elsken et al., 2019b). In NAS, a search algorithm is used to explore a search space of possible neural network architectures, aiming to find the best-performing architecture for a given task. Models originating from NAS techniques have already outperformed human-designed models on tasks such as image classification (Real et al., 2017; Zoph et al., 2018) and semantic segmentation (Chen et al., 2018). Based on the type of search algorithm used, NAS methods can be categorized into four groups: *(i)* Reinforcement Learning (RL) (Zoph & Le, 2017; Zoph et al., 2018; Baker et al., 2017), *(ii)* Evolutionary Algorithms (EA) (Real et al., 2017; Salimans et al., 2017), *(iii)* Bayesian Optimization (White et al., 2021), *(iv)* and Gradient-based methods (Liu et al., 2019). The latter has gained significant popularity in recent years due to its efficiency and ability to scale to large search spaces compared to the other methods.

Recently, the use of NAS for designing more efficient neural network architectures has been increasingly explored. Prior work, specifically on gradient-based NAS, often integrates hardware metrics as regularization terms into the optimization function (Cai et al., 2019; Wu et al., 2019; Wan et al., 2020). While this can help guide the search toward hardware-friendly architectures, these approaches often face several challenges. First, to be compatible with gradient-based optimization, the hardware metrics must be differentiable. Second, each regularization term requires appropriate weighting, introducing additional hyperparameters that demand careful tuning. Improper weighting can either overly penalize resource-intensive architectures (resulting to simpler models with suboptimal performance) or fail to enforce the hardware constraints (yielding architectures unsuitable for deployment on the target device). Consequently, these methods often require multiple search runs with varying weights assigned to the hardware-related terms until a feasible architecture is found, a process that can be both tedious and time-consuming.

To overcome these limitations, we introduce CONNAS, a novel hardware-aware, gradient-based NAS algorithm. Our work makes the following key contributions:

- **Direct constraint enforcement:** Unlike prior approaches that incorporate hardware metrics as regularization terms in the loss function, CONNAS modifies gradients with respect to architectural choices to directly enforce hardware constraints, effectively steering the search away from infeasible architectures.
- **No need for differentiable hardware metrics:** Our method eliminates the requirement for differentiable hardware metrics and the associated techniques needed to enforce differentiability.
- **Explicit constraint specification:** CONNAS allows for the explicit definition of hardware constraints without relying on hyperparameter tuning to balance the importance of hardware metrics.

We evaluate CONNAS on the NATS-Bench benchmark (Dong et al., 2021), demonstrating its ability to systematically find architectures that satisfy various hardware constraints while achieving performance close to the optimal architecture available within the search space, reaching as little as a $0.14\%$ difference in accuracy. Our experiments show that CONNAS significantly outperforms existing baseline methods, which often struggle to find feasible architectures under strict hardware constraints. Additionally, we validate CONNAS in a practical use case, where it successfully discovers high-performing architectures under strict hardware constraints (no more than a model size of 64kB and 18kB memory usage), consistently outperforming the best handcrafted architectures by up to $1.55\%$ accuracy.

## 2 RELATED WORK

**Hardware-aware NAS.** Over the years, the focus of NAS has shifted from discovering top-performing novel architectures to emphasizing computational efficiency. The design of neural networks has increasingly been guided by hardware-aware considerations, particularly in the context of deployment on resource-constrained devices. Early works primarily focused on optimizing hardware metrics which could be estimated based on the type of operations found in the model, such as the number of Floating Point Operations (FLOPs) (Xie et al., 2019; Zhou et al., 2018). However, other works have proposed to rely on real hardware metrics for more representative evaluation, which are generally obtained through look-up tables or regressors trained on real on-device benchmarks of various neural operations (Cai et al., 2019; Wu et al., 2019; Wan et al., 2020; Tan et al., 2019; Hu et al., 2020). Additionally, other contributions have a more specific focus on NAS techniques and search spaces tailored for ultra-constrained devices, such as microcontrollers (Lin et al., 2020; Liberis et al., 2021).

The most straightforward way to regularize hardware-related objectives is by incorporating hardware metrics as additional terms in the optimization function (Xie et al., 2019; Cai et al., 2019; Wu et al., 2019; Dong & Yang, 2019a; Wan et al., 2020; Bender et al., 2020; Hu et al., 2020). A weighting factor is then introduced to adjust the relative importance of these hardware metrics in comparison to the task-specific loss, effectively determining whether more constrained architectures should be favored. However, this fixed approach does not offer a direct mechanism for enforcing hardware constraints. Instead, it merely guides the search toward architectures with more favorable hardware metric values (typically simpler architectures with limited representational capacity) regardless of whether the current architecture actually satisfies the imposed constraints. In response,

Tan et al. (2019); Zhou et al. (2018); Dong & Yang (2019a); Bender et al. (2020) propose dynamic optimization functions where hardware-related terms are adjusted depending on whether the current architecture meets the hardware constraints. The rationale behind this approach is to find architectures that are close to the constraint boundary, which makes it possible to explore more complex architectures that still satisfy the constraints. This is in contrast to methods with a fixed loss term, which may limit the search to simpler architectures. Meanwhile, other techniques (Elsken et al., 2019a; Liberis et al., 2021), typically non-gradient-based methods, use a selection procedure based on hardware metric values, selecting only architectures that satisfy the hardware constraints for further optimization.

**Gradient-based NAS.** Gradient-based NAS techniques, originally introduced by DARTS (Liu et al., 2019), have been widely adopted in recent work to efficiently search for neural network architectures. In gradient-based NAS, all possible architectures are represented within an over-parameterized network, where each candidate path is associated with a continuous architecture weight $\alpha$. This formulation leads to a bi-level optimization problem, in which both the architecture $\alpha$ and its weights $w$ are jointly optimized using gradient descent. Traditionally, the loss function would be defined as the task-specific loss, such as cross-entropy for classification tasks. However, in hardware-aware NAS, the loss function is augmented with regularization terms to account for hardware metrics.

While hardware metrics for individual architectures are not differentiable, state-of-the-art techniques ensure that these metrics can be fully factorized over the architecture parameters. This enables expressing hardware metrics as a weighted sum of contributions from each candidate operation in the search space, thereby making them differentiable with respect to the architecture parameters. However, this approach requires knowledge of each operation's contribution to the overall hardware metric, which becomes computationally intractable in large search spaces. To address this, Xie et al. (2019) propose approximating the hardware-related terms using Monte Carlo estimation. Nevertheless, if the sampling distribution is poorly calibrated, the resulting estimates may be inaccurate.

**Constraint-aware Training.** Some recent NAS approaches incorporate hardware constraints directly into the optimization process, though this area remains relatively unexplored. HardCore-NAS (Nayman et al., 2021) addresses hard-constrained optimization by adapting the Frank-Wolfe algorithm to search for sub-networks within a pre-trained one-shot model. HDX (Hong et al., 2022) introduces hardware constraints by adding hardware metrics as regularization terms and modifying gradients based on differentiated hardware metrics. Similarly, Constraint Guided Gradient Descent (CGGD) (Van Baelen & Karsmakers, 2023), although not a NAS algorithm, enforces inequality constraints during training through gradient modification without penalty terms, proposing an update scheme that guides optimization toward weights satisfying the specified constraints. Building on these ideas, our work explores how such gradient-based enforcement can directly impose hardware constraints during the NAS search process, without requiring additional regularization terms or differentiable hardware metrics.

## 3 METHOD

### 3.1 PROBLEM FORMULATION

The search space $S$ is modeled as an over-parameterized neural network that encompasses all possible architectures. This is achieved by defining, for each layer in the network, a set of candidate operations (e.g., convolutions with varying kernel sizes) that can be used to construct a neural network. Each candidate operation is associated with a continuous architecture weight $\alpha$, which enables the use of gradient descent to jointly optimize both operation weights $w$ and the architecture weights $\alpha$. A final architecture $A$ can then be derived by selecting a subset of candidate operations based on the learned architecture weights $\alpha$. More formally, let the over-parameterized network $S$ be defined as a directed acyclic graph (DAG) with $L$ edges where each edge $e_l$ is associated with a set of candidate operations $\mathcal{O}_l = \{o_{l,1}, \ldots o_{l,N}\}$ with corresponding architecture weights $\boldsymbol{\alpha_l} = [\alpha_{l,1}, \ldots, \alpha_{l,N}]$ and operation weights $\boldsymbol{w_l} = [w_{l,1}, \ldots, w_{l,N}]$. An architecture $A \in S$ is derived by selecting, for each edge $e_l$, the operation with the highest architecture weight, i.e., $A = (o_{1,k_1}, \ldots, o_{L,k_L})$ where $k_l = \arg\max_k \alpha_{l,k}$. Let $c_k$ denote the $k$-th hardware metric function (e.g., number of parameters), and $b_k$ its upper bound. The set of hardware constraints is defined as $\mathcal{C} = \{(c_1, b_1), \ldots, (c_M, b_M)\}$, where each constraint is expressed as $c_k(A) \leq b_k$. The search process can then be formulated as

follows:

$$\underset{w,\alpha}{\arg\min}\, \mathcal{L}_{\text{task}}(w,\alpha) \quad \text{s.t.} \quad c_k(A) \leq b_k, \quad k = 1,\ldots,M \tag{1}$$

In other words, the goal is to find architecture weights $\alpha$ that yield an architecture $A(\alpha)$ minimizing the task-specific loss $\mathcal{L}_{\text{task}}$, while ensuring that the resulting architecture lies within the feasibility region $FR$, defined as the subset of architectures that satisfy all hardware constraints in $\mathcal{C}$. Formally, we aim to find $A(\alpha) \in FR$, or at least ensure that the sampled architectures $A^k$ generated during optimization converge to $FR$, i.e., $\lim_{k\to\infty} d(A^k, FR) = 0$.

The value for a hardware metric is obtained using $c_k(A)$, where $c_k$ may be implemented as a look-up table or a learned regressor. Unlike prior work on hardware-aware, gradient-based NAS (Cai et al., 2019; Wu et al., 2019; Wan et al., 2020; Dong & Yang, 2019a), CONNAS does not require differentiable hardware metrics with respect to the architecture weights $\alpha$.

### 3.2 GRADIENT UPDATE SCHEME

We adopt the gradient update scheme as proposed in CGGD (Van Baelen & Karsmakers, 2023) to enforce the hardware constraints $c_k(A) \leq b_k,\; k = 1,\ldots,M$. However, unlike CGGD, which aims to enforce constraints on the model's output predictions, we use this approach to manipulate the architecture weights $\alpha$ to steer the search process towards feasible architectures $A$. Specifically, each architecture weight $\alpha_{l,j}$ is updated as follows:

$$\alpha_{l,j} \leftarrow \alpha_{l,j} - \eta_\alpha \cdot (\nabla_{\alpha_{l,j}} \mathcal{L}_{\text{task}}(w,\alpha) + R \cdot dir_{\mathcal{C}}(\alpha_{l,j}) \cdot \max\left\{\left\|\nabla_{\alpha_{l,j}} \mathcal{L}_{\text{task}}(w,\alpha)\right\|, \epsilon\right\}) \tag{2}$$

Here, $R$ is a rescale factor that determines the strength of the constraints enforcement, and $\boldsymbol{dir_{\mathcal{C}}(\alpha_l)} = [dir_{\mathcal{C}}(\alpha_{l,1}),\ldots,dir_{\mathcal{C}}(\alpha_{l,N})]$ is a unit vector indicating the direction in which the architecture weights should be modified to satisfy the constraints. An arbitrarily small constant $\epsilon$ is added to allow adjustments even when the loss gradient is zero.

### 3.3 GRADIENT DIRECTION

The choice of $R$ and $\boldsymbol{dir_{\mathcal{C}}(\alpha_l)}$ is crucial, as it determines whether the architecture weights are modified in a way that effectively reduces the hardware constraint violations, and ultimately leads to convergence to a feasible architecture. Van Baelen & Karsmakers (2023) proved that convergence towards the $FR$ is guaranteed if $R > 1$ and $\boldsymbol{dir_{\mathcal{C}}(\alpha_l)}$ is chosen to be the shortest path with respect to the Euclidean distance to the feasibility region. They also showed that performance is not highly sensitive to the exact value of $R$, as long as it is strictly larger than 1. In other words, this eliminates the need for additional hyperparameter tuning, unlike prior work that relies on weighted regularization terms.

However, computing the exact shortest path is often intractable because it requires knowing the hardware metrics for all possible architectures in the search space, or at least that these metrics are fully factorizable across all edges. To address this, we propose a heuristic approach to approximate the direction of the global shortest path by calculating the shortest path for each edge $e_l$ independently. However, this means that the same convergence guarantees as in Van Baelen & Karsmakers (2023) do not strictly hold. Nevertheless, our experiments (Section 4.1) demonstrate that this heuristic is effective in practice.

For each edge $e_l$, we first evaluate the hardware metric values of all candidate operations $\mathcal{O}_l$ under the current architecture configuration. Then, we construct a set of candidate architectures $\mathcal{A}_l = \{A_{l,1},\ldots,A_{l,N}\}$ by replacing the $l$-th operation with $o_{l,j}$ while keeping the rest fixed:

$$A_{l,j} = (o_{1,k_1},\ldots,o_{l,j},\ldots,o_{L,k_L}) \tag{3}$$

We then evaluate $c_k(A_{l,j})$ for each candidate and constraint $(c_k, b_k) \in \mathcal{C}$ in order to determine $\boldsymbol{dir_{\mathcal{C}}^k(\alpha_l)}$, which is the direction to the $FR$ for constraint $c_k(A_{l,j}) \leq b_k$. Based on whether the candidate architectures $\mathcal{A}_l$ satisfy the hardware constraint, we distinguish three cases:

**1. All candidates on $e_l$ satisfy the hardware constraint** ($\forall j \in \{1,\ldots,N\},\; c_k(A_{l,j}) \leq b_k$). In this case, we set $\boldsymbol{dir_{\mathcal{C}}^k(\alpha_l)} = 0$ for all candidates, as no modification is needed to satisfy the constraint for this edge.

**2. Some candidates on $e_l$ violate the hardware constraint** ($\exists j,\; c_k(A_{l,j}) > b_k$). In this situation, the direction is computed by aggregating the unit vectors that point away from the candidates that

violate the constraint, while pointing towards those that satisfy it. More specifically, we first identify the set of candidates that satisfy the constraint:

$$\mathcal{F} = \{j \mid c_k(A_{l,j}) \leq b_k\} \tag{4}$$

For each pair $(j, m)$ where $j \in \mathcal{F}$ and $m \notin \mathcal{F}$, we compute the unit vector that points from candidate $m$ to candidate $j$, as shown in Figure 1a:

$$\boldsymbol{u_{j,m}} = [\underbrace{0, \ldots, 0}_{j-1}, 1, \underbrace{0, \ldots, 0}_{m-j-1}, -1, \underbrace{0, \ldots, 0}_{N-m}]/\sqrt{2} \tag{5}$$

Then, the final direction $\boldsymbol{dir_c^k(\alpha_l)}$ is determined by summing these unit vectors and normalizing the result to a unit vector:

$$\boldsymbol{dir_c^k(\alpha_l)} = \sum_{j \in \mathcal{F}} \sum_{m \notin \mathcal{F}} \boldsymbol{u_{j,m}} / \left\| \sum_{j \in \mathcal{F}} \sum_{m \notin \mathcal{F}} \boldsymbol{u_{j,m}} \right\| \tag{6}$$

**3. No candidate on $e_l$ satisfies the hardware constraint** ($\forall j \in \{1, \ldots, N\}$, $c_k(A_{l,j}) > b_k$). In this case, the direction is computed in such that candidates with higher hardware metric values are penalized while those with lower values are rewarded. First, the candidates are ranked based on their hardware metric values. Then, for each ranked candidate, we compute unit vectors $\boldsymbol{u_{j,m}}$ (as defined in Equation 5) that point from candidates with higher hardware metric values to those with lower ones. Specifically:

- First, we compute the unit vectors between the candidate with the highest hardware metric value and all other candidates.

- Then, we compute the unit vectors between the two candidates with the highest hardware metric values and all others.

- This process is repeated until all candidates, except the one with the lowest hardware metric value, have been considered, as shown in Figure 1b. Finally, the unit vectors are aggregated in the same manner as in Equation 6.

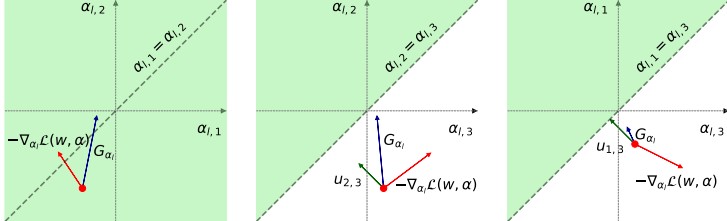

(a) Both $A_{l,1}$ and $A_{l,2}$ satisfy the constraint, $A_{l,3}$ violates it, so the resulting gradient $G_{\alpha_l}$ is decreased for $\alpha_{l,3}$ and increased for $\alpha_{l,1}$ and $\alpha_{l,2}$.

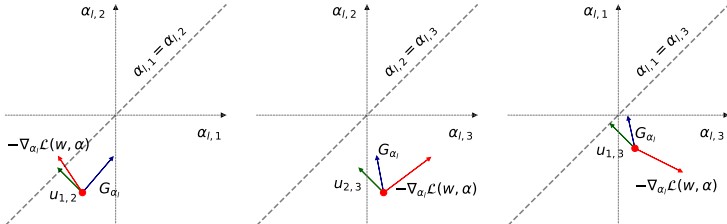

(b) No candidate for $e_j$ satisfies the constraint, so the resulting gradient $G_{\alpha_l}$ is increased / decreased based on the ranked hardware metric values of each candidate.

Figure 1: Architecture space for a single edge $e_l$ with three candidate operations, parameterized by $\alpha_{l,1}, \alpha_{l,2}, \alpha_{l,3}$, where $c_k(A_{l,1}) < c_k(A_{l,2}) < c_k(A_{l,3})$. Each subfigure shows a projection onto the plane defined by two architecture weights. The shaded area represents the feasibility region $FR$ for $c_k$. $\boldsymbol{G_{\alpha_l}}$ is the gradient after modification projected into each plane.

---

**Algorithm 1** Training procedure under hardware constraints

---

Input: Search space $S$, current architecture $A$ derived from $\alpha$, hardware constraints $\mathcal{C}$, rescale factor $R$, epsilon $\epsilon$, learning rates $\eta_w, \eta_\alpha$, loss function $\mathcal{L}_{\text{task}}$, batch of data $(X, y)$
$w \leftarrow w - \eta_w \cdot \nabla_w \mathcal{L}_{\text{task}}(w, \alpha)$ {Update operation weights}
$\boldsymbol{G_\alpha} = \nabla_\alpha \mathcal{L}_{\text{task}}(w, \alpha)$ {Store gradients w.r.t. architecture weights}
$dirs_c = [\,]$ {For each unsatisfied hardware constraint, compute direction to feasibility region}
**for** $(c_k, b_k) \in \mathcal{C}$ **do**
   **if** $c_k(A) > b_k$ **then**
      Append $calculate\_gradient\_direction(A, c_k)$ to $dirs_c$
   **end if**
**end for**
$\boldsymbol{dir_{\mathcal{C}}(\alpha)} = \sum_{\boldsymbol{dir_c^k} \in dirs_c} \boldsymbol{dir_c^k} / \left\| \sum_{\boldsymbol{dir_c^k} \in dirs_c} \boldsymbol{dir_c^k} \right\|$ {Aggregate directions into single unit vector}
$\boldsymbol{G_\alpha} \leftarrow \boldsymbol{G_\alpha} + R \cdot \boldsymbol{dir_{\mathcal{C}}(\alpha)} \cdot \max\{\|\boldsymbol{G_\alpha}\|, \epsilon\}$ {Modify gradient toward the feasibility region}
$\alpha \leftarrow \alpha - \eta_\alpha \boldsymbol{G_\alpha}$ {Update architecture weights}

---

## 3.4 Constraint-guided Hardware-aware NAS

We refer to Algorithm 1 for the modified training procedure. Within each batch, the optimization of operation weights $w$ and architecture weights $\alpha$ is performed in an interleaved manner, as proposed in prior work (Cai et al., 2019; Wu et al., 2019; Wan et al., 2020; Dong & Yang, 2019a;b). After optimizing $w$, the gradient $\nabla_\alpha \mathcal{L}_{\text{task}}(w, \alpha)$ is computed, but instead of immediately updating $\alpha$, it is stored for later modification. Then, for each hardware metric $c_k$ and its associated constraint $c_k(A) \leq b_k$, the current architecture $A$ is checked for violations. If a constraint is violated, the direction for modification $\boldsymbol{dir_c^k(\alpha)}$ is computed as described in Section 3.3. All directions are summed and normalized to a unit vector to obtain $\boldsymbol{dir_{\mathcal{C}}(\alpha)}$. Finally, the stored gradient is modified, as explained in Section 3.2, and used to update the architecture weights $\alpha$.

## 4 Experiments and Results

We evaluate the effectiveness of CONNAS on two benchmark tasks: the NATS-Bench topology and size search spaces (Section 4.1). In addition, we apply our method to a practical use case involving edge-based condition monitoring of induction motors, where the goal is to discover architectures suitable for deployment across diverse hardware configurations (Section 4.2).

### 4.1 Experiments on NATS-Bench

NATS-Bench, proposed by Dong et al. (2021), is a unified benchmark for NAS designed for image classification tasks, including two search spaces: the topology search space and the size search space. The topology search space is a cell-based search space, which contains 15,625 unique architectures, while size search space is a layer-wise search space, consisting of 32,768 unique architectures. We refer to Dong et al. (2021) for a detailed description of the search spaces. A key advantage of NATS-Bench over other NAS benchmarks is that it provides test accuracies for all architectures across both search spaces on CIFAR-10, CIFAR-100, and ImageNet16-120. This enables rapid evaluation without the need for retraining. Additionally, we combine both real on-device measurements and proxy-based hardware metrics to define hardware constraints for our experiments. Specifically, we use on-device measurements of inference latency and energy usage on an NVIDIA Jetson TX2 Edge GPU (NVIDIA Inc.) for the topology search space, as provided by HW-NAS-Bench (Li et al., 2021). For the size search space, we incorporate widely used proxy-based metrics, including the number of parameters, FLOPs, and peak memory usage, all of which can be computed directly from the architecture definition. A detailed description of how these proxy-based hardware metrics are calculated is provided in Appendix A. Importantly, our approach is flexible and can integrate any other hardware metric, regardless of how it is obtained or computed.

We use a Gumbel-Softmax (Jang et al., 2017), as proposed in Dong & Yang (2019a); Wu et al. (2019); Wan et al. (2020), to relax the categorical distribution of the candidate operations $\mathcal{O}_l$. At each edge $e_l$, the output $y_l$ is computed as a weighted sum of the outputs produced by all candidate

operations:

$$\boldsymbol{y_l} = \sum_{j}^{N} \frac{\exp((\log(\alpha_{l,j}) + g_{l,j})/\tau)}{\sum_{j}^{N} \exp((\log(\alpha_{l,j}) + g_{l,j})/\tau)} \cdot o_{l,j}(\boldsymbol{x_l}) \qquad (7)$$

where $g_{l,j} \sim \text{Gumbel}(0,1)$ and temperature $\tau$ controls the smoothness of the distribution. A higher temperature is used at the beginning of the search to encourage exploration, making the distribution closer to uniform. As the search progresses, the temperature is gradually lowered, making the distribution sharper and closer to an $argmax$, which promotes exploitation. It is worth noting that CONNAS is agnostic to the specific relaxation method used. Other approaches, such as proposed by DARTS (Liu et al., 2019) or ProxylessNAS (Cai et al., 2019), although not tested here, could also be applied.

We run CONNAS on each search space and corresponding set of hardware constraints for a total of 150 epochs. During the first 100 epochs, the temperature $\tau$ is linearly annealed from 10 to 0.1. The rescale factor $R$ is set to 1.2, though additional experiments show that the performance is not very sensitive to this choice (see Appendix B). Among the final 50 epochs, we select the architecture with the lowest validation loss that satisfies the hardware constraints. Finally, the performance of the selected architecture is obtained from the test accuracies provided by NATS-Bench. Each experiment is repeated 5 times, as in Dong et al. (2021). The mean and standard deviation of the results are reported in Table 2 and Table 3 for the topology and size search spaces, respectively. Since NATS-Bench provides performance data for each architecture in the search space, we also report the relative error between parentheses to the optimal architecture satisfying the hardware constraints. Results of individual runs are shown in Figure 2.

We further compare CONNAS against several hardware-aware, gradient-based baselines listed in Table 1. We adopt the same experimental setup and training procedure for these baselines to ensure a fair comparison. More training details are provided in Appendix E.

CONNAS consistently identifies architectures that satisfy the specified hardware constraints while maintaining strong predictive performance. The resulting architectures exhibit relative errors of at most $-1.18\%$, $-4.10\%$, and $-6.45\%$ compared to the optimal feasible solution on CIFAR-10, CIFAR-100, and ImageNet16-120, respectively. Compared to the baseline methods using the weighting factors listed in Table 1, CONNAS achieves substantially better results than Proxyless-NAS and HDX, and a performance comparable to TF-NAS. In contrast, FBNet and TAS do not consistently produce valid architectures under all constraint configurations. Additional experiments under stricter hardware constraints, presented in Appendix C, further demonstrate the effectiveness of CONNAS, which continues to find valid architectures and generally outperforms all baselines.

It is important to note that the weighting factors used for the baselines may not be optimal under the given constraints, and further tuning could potentially improve their performance. Nevertheless, this underscores a key advantage of CONNAS: it eliminates the need for extensive hyperparameter tuning to balance hardware metrics, thereby eliminating the need to rerun experiments multiple times to find an appropriate weighting factor. To illustrate this, we conduct additional experiments with alternative weighting factors for the baselines (see Appendix D) and an ablation study on the rescale factor $R$ for CONNAS (see Appendix B). Both analyses confirm that CONNAS is more robust to variations in the rescale factor, whereas baseline methods exhibit higher sensitivity to weighting factors, requiring careful tuning for each hardware constraint setting.

## 4.2 PRACTICAL USE CASE: CONDITION MONITORING OF INDUCTION MOTORS

Condition monitoring of industrial assets has received increasing interest over the years (Surucu et al., 2023), preventing unexpected failures and costly downtimes. By leveraging edge machine learning, data from these assets can be captured and processed locally to detect faults in real-time. To demonstrate the applicability of our work, we validate our approach through a use case focused on condition monitoring of induction motors, aiming to detect eccentricity faults using a neural network deployed on an edge device. An eccentricity fault occurs when the rotor is not perfectly centered within the stator, resulting in an uneven air gap, causing an unbalanced magnetic pull (Desenfans et al., 2024). The prediction task involves classifying the type of eccentricity, specifically distinguishing between no eccentricity fault, static eccentricity (where the rotor remains consistently off-center relative to the stator), dynamic eccentricity (where the rotor's off-center position rotates

Table 1: **Overview of baseline methods used for comparison.** Here, $c'_k(\alpha)$ is the normalized form of $c_k(\alpha)$, following the same procedure described in the original work. The term $b_k$ represents the upper bound associated with $c_k$. $\lambda_{\text{hardware}}$ is a weighting factor. We use $\lambda_{\text{hardware}} = 1$ for ProxylessNAS, TAS, TF-NAS and HDX, and $\lambda_{\text{hardware}} = 0.6$ for FBNet. Additional experiments with varying weighting factors for $\lambda_{\text{hardware}}$ are provided in Appendix D.

| Name | Regularization |
|---|---|
| ProxylessNAS (Cai et al., 2019) | $\mathcal{L}(w,\alpha) = \mathcal{L}_{\text{task}}(w,\alpha) + (\lambda_{\text{hardware}}/M) \cdot \sum_{k=1}^{M} c'_k(\alpha)$ |
| FBNet (Wu et al., 2019) | $\mathcal{L}(w,\alpha) = \mathcal{L}_{\text{task}}(w,\alpha) \cdot \prod_{k=1}^{M} c'_k(\log \alpha)^{\lambda_{\text{hardware}}}$ |
| TAS (Dong & Yang, 2019a) | $\mathcal{L}(w,\alpha) = \mathcal{L}_{\text{task}}(w,\alpha) + (\lambda_{\text{hardware}}/M) \cdot \sum_{k=1}^{M} h_j(\alpha)$ 
 where $h_j(\alpha) = \begin{cases} c'_k(\alpha), & \text{if } c_k(A) > b_k \\ -c'_k(\alpha), & \text{if } c_k(A) < b_k \\ 0, & \text{otherwise} \end{cases}$ |
| TF-NAS (Hu et al., 2020) | $\mathcal{L}(w,\alpha) = \mathcal{L}_{\text{task}}(w,\alpha) + (\lambda_{\text{hardware}}/M) \cdot \sum_{k=1}^{M} \max(\frac{c_k(\alpha)}{b_k} - 1, 0)$ |
| HDX (Hong et al., 2022) | $G_\alpha = \begin{cases} \nabla_\alpha \mathcal{L}(w,\alpha), & \text{if } c_k(\alpha) \leq b_k \\ m_\alpha + \nabla_\alpha \mathcal{L}(w,\alpha), & \text{otherwise} \end{cases}$ 
 where $\mathcal{L}(w,\alpha) = \mathcal{L}_{\text{task}}(w,\alpha) + (\lambda_{\text{hardware}}/M) \cdot \sum_{k=1}^{M} c'_k(\alpha)$ |

Table 2: **NATS-Bench topology search space classification results.** Test accuracy (mean ± std over 5 runs) is reported under hardware constraints, using on-device measurements of inference latency and energy usage on an NVIDIA Jetson TX2 Edge GPU. The constraints are chosen so that approximately 50% of architectures in the search space satisfy the constraints. The relative error compared to the optimal architecture satisfying the hardware constraints is reported between parentheses. Best results for each constraint and dataset are highlighted in bold. Performance of runs that do not always satisfy the constraints are reported in gray.

| Method | Top-1 accuracy (%) | | | Runs |
|---|---|---|---|---|
| | CIFAR-10 | CIFAR-100 | ImageNet16-120 | satisfied |
| $c_{\text{ts,latency}}$ : latency $\leq 5.31ms$ *(7814 architectures satisfied)* | | | | |
| **ConNAS** | $93.18$ ±0.09 *(-1.08)* | **$70.12$ ±0.16 *(-3.04)*** | **$40.59$ ±1.20 *(-6.01)*** | 5 / 5 |
| ProxylessNAS | $92.29$ ±0.00 *(-1.97)* | $67.48$ ±0.00 *(-5.68)* | $39.40$ ±0.00 *(-7.20)* | 5 / 5 |
| FBNet | $93.18$ ±0.12 *(-1.08)* | $70.00$ ±0.08 *(-3.16)* | $39.71$ ±0.06 *(-6.89)* | 5 / 5 |
| TAS | $93.67$ ±0.00 | $70.91$ ±0.00 | $41.02$ ±0.00 | 0 / 5 |
| TF-NAS | **$93.27$ ±0.00 *(-1.00)*** | $70.06$ ±0.00 *(-3.10)* | $39.75$ ±0.00 *(-6.85)* | 5 / 5 |
| HDX | $75.83$ ±36.80 *(-18.43)* | $54.18$ ±29.73 *(-18.98)* | $31.69$ ±17.25 *(-14.91)* | 5 / 5 |
| $c_{\text{ts,energy}}$ : energy usage $\leq 23.95mJ$ *(7814 architectures satisfied)* | | | | |
| ConNAS | $92.97$ ±0.54 *(-1.18)* | $69.51$ ±1.48 *(-2.43)* | **$40.32$ ±1.56 *(-6.45)*** | 5 / 5 |
| ProxylessNAS | $92.29$ ±0.00 *(-1.86)* | $67.48$ ±0.00 *(-4.46)* | $39.40$ ±0.00 *(-7.37)* | 5 / 5 |
| **FBNet** | **$93.05$ ±0.00 *(-1.10)*** | **$69.91$ ±0.00 *(-2.03)*** | $39.64$ ±0.00 *(-7.12)* | 5 / 5 |
| TAS | $93.67$ ±0.00 | $70.91$ ±0.00 | $41.02$ ±0.00 | 0 / 5 |
| TF-NAS | $93.01$ ±0.57 *(-1.14)* | $69.48$ ±1.29 *(-2.46)* | $39.62$ ±0.30 *(-7.15)* | 5 / 5 |
| HDX | $92.29$ ±0.00 *(-1.86)* | $67.48$ ±0.00 *(-4.46)* | $39.40$ ±0.00 *(-7.37)* | 5 / 5 |
| $c_{\text{ts,comb}}$ : latency $\leq 5.31ms \wedge$ energy usage $\leq 23.95mJ$ *(7586 architectures satisfied)* | | | | |
| **ConNAS** | $93.14$ ±0.05 *(-1.01)* | **$70.21$ ±0.17 *(-1.73)*** | **$41.45$ ±1.01 *(-5.08)*** | 5 / 5 |
| ProxylessNAS | $92.29$ ±0.00 *(-1.86)* | $67.48$ ±0.00 *(-4.46)* | $39.40$ ±0.00 *(-7.13)* | 5 / 5 |
| FBNet | $92.01$ ±0.00 *(-2.14)* | $67.07$ ±0.00 *(-4.87)* | $39.19$ ±0.00 *(-7.34)* | 5 / 5 |
| TAS | $93.67$ ±0.00 | $70.91$ ±0.00 | $41.02$ ±0.00 | 0 / 5 |
| TF-NAS | **$93.27$ ±0.00 *(-0.89)*** | $70.06$ ±0.00 *(-1.88)* | $39.75$ ±0.00 *(-6.78)* | 5 / 5 |
| HDX | $92.29$ ±0.00 *(-1.86)* | $67.48$ ±0.00 *(-4.46)* | $39.40$ ±0.00 *(-7.13)* | 5 / 5 |

over time relative to the stator), and mixed eccentricity (a combination of static and dynamic eccentricity).

The goal in this use case is to discover neural network architectures that are deployable across a wide range of microcontrollers with diverse hardware capabilities. To define realistic deployment constraints, we base our hardware constraints on 6 hand-crafted architectures (listed in Appendix G), each having different resource requirements. We construct a search space based on 1D convolutions consisting of 8 layers. Each layer has varying configurations in terms of the number of filters, convolution types, and the inclusion of skip connections, resulting in a total of 1.94 billion unique

Table 3: **NATS-Bench size search space classification results.** Test accuracy (mean ± std over 5 runs) is reported under hardware constraints, including the number of parameters, number of FLOPs, and peak memory usage. The constraints are chosen such that about 50% of the architectures in each search space meet them. The relative error compared to the optimal architecture satisfying the hardware constraints is reported between parentheses. Best results for each constraint and dataset are highlighted in bold. Performance of runs that do not always satisfy the constraints are reported in gray.

| Method | Top-1 accuracy (%) | | | Runs |
| --- | --- | --- | --- | --- |
| | CIFAR-10 | CIFAR-100 | ImageNet16-120 | satisfied |
| $c_{\text{ss,param}}$ : #parameters $\leq 261650$ *(16385 architectures satisfied)* | | | | |
| **ConNAS** | $91.74 _{\pm 0.29}$ *(-0.66)* | **$66.22 _{\pm 0.47}$ *(-2.70)*** | **$39.77 _{\pm 0.77}$ *(-2.26)*** | 5 / 5 |
| ProxylessNAS | $85.87 _{\pm 1.16}$ *(-6.66)* | $51.11 _{\pm 2.35}$ *(-17.81)* | $25.27 _{\pm 2.13}$ *(-16.76)* | 5 / 5 |
| FBNet | $93.22 _{\pm 0.20}$ | $69.43 _{\pm 0.22}$ | $45.68 _{\pm 0.37}$ | 0 / 5 |
| TAS | $92.25 _{\pm 0.02}$ | $67.34 _{\pm 0.54}$ | $41.89 _{\pm 0.33}$ | 0 / 5 |
| TF-NAS | **$91.78 _{\pm 0.00}$ *(-0.75)*** | $63.64 _{\pm 0.00}$ *(-5.28)* | $38.00 _{\pm 0.00}$ *(-4.03)* | 5 / 5 |
| HDX | $86.10 _{\pm 1.28}$ *(-6.43)* | $51.71 _{\pm 2.88}$ *(-17.21)* | $25.36 _{\pm 2.42}$ *(-16.67)* | 5 / 5 |
| $c_{\text{ss,mem}}$ : peak memory usage $\leq 655kB$ *(20480 architectures satisfied)* | | | | |
| ConNAS | **$93.28 _{\pm 0.15}$ *(-0.14)*** | $69.70 _{\pm 1.02}$ *(-1.16)* | $44.58 _{\pm 0.43}$ *(-1.35)* | 5 / 5 |
| ProxylessNAS | $91.33 _{\pm 0.19}$ *(-2.09)* | $68.22 _{\pm 0.28}$ *(-2.64)* | $38.49 _{\pm 0.03}$ *(-7.45)* | 5 / 5 |
| FBNet | $93.16 _{\pm 0.29}$ | $69.69 _{\pm 0.05}$ | $45.53 _{\pm 0.87}$ | 1 / 5 |
| TAS | $92.93 _{\pm 0.19}$ *(-0.49)* | $70.10 _{\pm 0.32}$ *(-0.76)* | $44.98 _{\pm 0.59}$ *(-0.95)* | 5 / 5 |
| **TF-NAS** | $92.99 _{\pm 0.10}$ *(-0.43)* | **$70.21 _{\pm 0.16}$ *(-0.65)*** | **$45.25 _{\pm 0.27}$ *(-0.69)*** | 5 / 5 |
| HDX | $91.59 _{\pm 0.19}$ *(-1.83)* | $68.03 _{\pm 0.29}$ *(-2.83)* | $38.41 _{\pm 0.07}$ *(-7.53)* | 5 / 5 |
| $c_{\text{ss,flops}}$ : #FLOPs $\leq 1196M$ *(16385 architectures satisfied)* | | | | |
| **ConNAS** | **$91.71 _{\pm 0.28}$ *(-1.09)*** | $65.44 _{\pm 0.80}$ *(-4.10)* | **$38.33 _{\pm 0.36}$ *(-5.13)*** | 5 / 5 |
| ProxylessNAS | $89.55 _{\pm 0.22}$ *(-3.25)* | $63.51 _{\pm 0.67}$ *(-6.03)* | $32.43 _{\pm 0.84}$ *(-11.03)* | 5 / 5 |
| FBNet | $93.02 _{\pm 0.09}$ | $69.28 _{\pm 0.36}$ | $45.04 _{\pm 0.60}$ | 0 / 5 |
| TAS | $92.15 _{\pm 0.19}$ | $68.66 _{\pm 0.52}$ | $42.11 _{\pm 0.82}$ | 3 / 5 |
| TF-NAS | $91.10 _{\pm 0.25}$ *(-1.70)* | **$66.56 _{\pm 1.12}$ *(-2.98)*** | $37.41 _{\pm 0.94}$ *(-6.06)* | 5 / 5 |
| HDX | $84.43 _{\pm 0.99}$ *(-8.37)* | $49.02 _{\pm 0.85}$ *(-20.52)* | $23.17 _{\pm 1.51}$ *(-20.29)* | 5 / 5 |
| $c_{\text{ss,comb}}$ : #parameters $\leq 261650 \wedge$ peak memory usage $\leq 655kB \wedge$ #FLOPs $\leq 1196M$ *(13956 architectures satisfied)* | | | | |
| **ConNAS** | **$91.72 _{\pm 0.35}$ *(-0.65)*** | **$65.95 _{\pm 0.52}$ *(-2.97)*** | **$39.52 _{\pm 0.46}$ *(-2.25)*** | 5 / 5 |
| ProxylessNAS | $84.12 _{\pm 0.28}$ *(-8.25)* | $49.12 _{\pm 1.07}$ *(-19.80)* | $22.64 _{\pm 0.31}$ *(-19.13)* | 5 / 5 |
| FBNet | $92.59 _{\pm 0.32}$ | $67.92 _{\pm 0.64}$ | $43.09 _{\pm 1.12}$ | 0 / 5 |
| TAS | $92.50 _{\pm 0.00}$ | $67.02 _{\pm 0.00}$ | $41.30 _{\pm 0.00}$ | 0 / 5 |
| TF-NAS | $90.93 _{\pm 0.21}$ *(-1.44)* | $65.41 _{\pm 0.37}$ *(-3.51)* | $37.03 _{\pm 0.46}$ *(-4.73)* | 5 / 5 |
| HDX | $83.99 _{\pm 0.00}$ *(-8.38)* | $48.64 _{\pm 0.00}$ *(-20.28)* | $22.50 _{\pm 0.00}$ *(-19.27)* | 5 / 5 |

architectures. The detailed description of the search space is provided in Appendix F. The dataset contains 31,920 instances, split into 25,530 for training and 6,390 for testing. Each instance consists of 256 time steps sampled at 5 kHz, with eight features: *stator currents* (3), *phase voltages* (3), *rotor speed* (1), and *rotor angle* (1). We use 80% of the training set to train the supernet, while the remaining 20% is used as a validation set for architecture selection. The training and selection procedure follows the same approach outlined in Section 4.1. The selected architectures are retrained from scratch using five-fold cross-validation over 200 epochs. Their performance is subsequently evaluated on the test set. We compare the best architectures found by CONNAS with hand-crafted models.

Table 4 compares the performance and resource consumption of architectures discovered by CON-NAS with manually designed baselines. CONNAS consistently identifies architectures that outperform handcrafted ones in terms of accuracy under specific hardware constraints. For example, under constraint $c_1$, CONNAS discovers architectures that achieve an accuracy of 97.07%, which is 1.55% higher than Conv1D-Reg, the best manually designed model under the same constraint. At $c_4$, the discovered architecture reaches 94.37% accuracy, compared to 93.93% for Conv1D-Reg-Min.

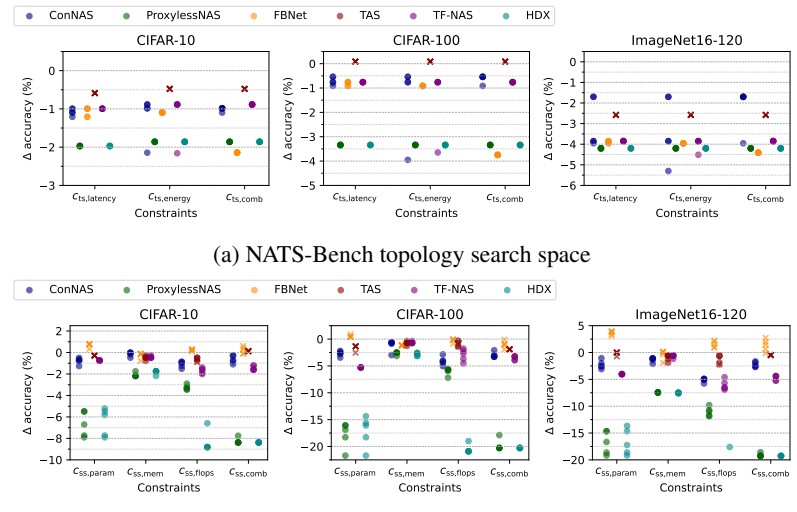

(a) NATS-Bench topology search space

(b) NATS-Bench size search space

Figure 2: **Performance comparison of CONNAS against baseline methods.** Performance is measured as the difference in test accuracy (%) between the architecture found by each method and the optimal architecture satisfying the hardware constraints. Each point represents a run, runs violating hardware constraints are marked with a cross.

Table 4: **Classification performance and resource usage of models discovered by CONNAS under various hardware constraints, compared to handcrafted baseline models.** The constraints $c_1$, $c_2$, $c_3$, and $c_4$ are defined as follows: $c_1$: model size $\leq$ 2MB, peak memory usage $\leq$ 640kB; $c_2$: model size $\leq$ 256kB, peak memory usage $\leq$ 112kB; $c_3$: model size $\leq$ 128kB, peak memory usage $\leq$ 36kB; $c_4$: model size $\leq$ 64kB, peak memory usage $\leq$ 18kB.

| Name | Top-1 accuracy (%) | Satisfied constraints | | | | Model size (bytes) | Peak memory usage (bytes) |
|---|---|---|---|---|---|---|---|
| | | $c_1$ | $c_2$ | $c_3$ | $c_4$ | | |
| ConNAS (unconstrained) | 97.36 ±0.27 | | | | | 6.06M ± 1.61M | 524k ± 0.00 |
| ConNAS + $c_1$ | 97.09 ±0.47 | ✓ | | | | 1.93M ± 13.2k | 524k ± 0.00 |
| ConNAS + $c_2$ | 96.17 ±0.84 | ✓ | ✓ | | | 122k ± 30.7k | 69.6k ± 0.00k |
| ConNAS + $c_3$ | 94.96 ±0.69 | ✓ | ✓ | ✓ | | 26.0k ± 13.5k | 16.8k ± 10.1k |
| ConNAS + $c_4$ | 94.37 ±1.94 | ✓ | ✓ | ✓ | ✓ | 8.69k ± 2.12k | 12.3k ± 0.00 |
| Conv1D-Reg | 95.54 ±0.29 | ✓ | | | | 135k | 16.4k |
| Conv1D-DS | 93.07 ±0.35 | ✓ | ✓ | ✓ | ✓ | 51.3k | 16.4k |
| Conv1D-Reg-Max | 93.31 ±1.86 | | | | | 22.1M | 524k |
| Conv1D-DS-Max | 96.72 ±0.24 | | | | | 7.50M | 524k |
| Conv1D-Reg-Min | 90.15 ±4.33 | ✓ | ✓ | ✓ | ✓ | 3.47k | 12.3k |
| Conv1D-DS-Min | 93.93 ±1.64 | ✓ | ✓ | ✓ | ✓ | 2.32k | 12.3k |

## 5 CONCLUSION

We introduced CONNAS, a novel hardware-aware, gradient-based NAS technique that explicitly enforces hardware constraints during the search process. Unlike prior approaches that rely on weighted loss terms for hardware-aware regularization, CONNAS directly modifies the gradients of architecture parameters when hardware constraints are violated, effectively steering the search toward hardware-feasible solutions. Experiments on NATS-Bench demonstrate that CONNAS consistently discovers architectures that satisfy various hardware constraints, achieving performance close to the optimal and overall outperforming existing baseline methods in both performance and compliance with hardware constraints. Furthermore, we validated CONNAS in a practical use case, where it successfully identified high-performing architectures within tight resource budgets. These results highlight CONNAS as a promising approach for the automatic design of deep learning models for deployment in resource-constrained environments.

ACKNOWLEDGMENTS

The resources and services used in this work were provided by the VSC (Flemish Supercomputer Center), funded by the Research Foundation - Flanders (FWO) and the Flemish Government.

This research was supported by the Research Foundation - Flanders (FWO) under the SBO grant program 1SH9Y24N.

This research was supported by Internal Funds KU Leuven (C3/23/047, C3 Retrokit).

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

# A    COMPUTATION HARDWARE METRICS

In this work, we focus on three hardware metrics: the number of parameters, the number of FLOPs, and the peak memory usage during inference. The following subsections describe how each of these metrics is computed. It is important to note that our proposed method is not limited to these metrics and can be applied to any hardware metric, whether derived analytically, measured through profiling, or predicted by learned models.

## A.1    NUMBER OF PARAMETERS

The total number of parameters in a candidate architecture is computed by summing the number of parameters of each individual layer. To estimate the model size in bytes, we multiply the total number of parameters by 4, assuming 32-bit floating-point representation.

### A.1.1    CONVOLUTIONAL BLOCK

A convolutional block consists of a convolutional layer, followed by a batch normalization and a ReLU activation. We assume that batch normalization is fused into the convolutional layer during inference.

*Regular Convolution:*
$$\text{params} = (C_{in} \times k_h \times k_w + 1) \times C_{out} \tag{8}$$

where $C_{in}$ and $C_{out}$ are the input and output channels, $k_h$ and $k_w$ are kernel dimensions, and the $+1$ accounts for the bias term.

*Depthwise Separable Convolution:*
$$\text{params} = (C_{in} \times k_h \times k_w + 1) + (C_{in} \times C_{out} + 1) \tag{9}$$

where the first term corresponds to the depthwise convolution and the second to the pointwise convolution.

## A.2    FULLY CONNECTED LAYER

$$\text{params} = (N_{in} + 1) \times N_{out} \tag{10}$$

where $N_{in}$ and $N_{out}$ are the input and output features, respectively, and the $+1$ accounts for the bias term.

## A.3    NUMBER OF FLOPS

Similar to the parameter count, the total number of FLOPs is computed by summing the FLOPs of each individual layer.

### A.3.1    CONVOLUTIONAL LAYER

$$\text{FLOPs} = 2 \times (C_{in} \times k_h \times k_w + 1) \times H_{out} \times W_{out} \times C_{out} \tag{11}$$

Here, $H_{out}$ and $W_{out}$ are the output feature map dimensions.

### A.3.2    RELU

$$\text{FLOPs} = H_{out} \times W_{out} \times C_{out} \tag{12}$$

### A.3.3    LINEAR LAYER

$$\text{FLOPs} = 2 \times (N_{in} + 1) \times N_{out} + N_{out} \tag{13}$$

### A.3.4    GLOBAL POOLING LAYER

$$\text{FLOPs} = (k_h \times k_w + 1) \times H_{out} \times W_{out} \times C_{out} \tag{14}$$

### A.4 PEAK MEMORY USAGE

Peak memory usage during inference is estimated by computing the combined size of input and output feature maps for each layer. The maximum of these values across all layers is taken as the peak memory usage.

# B  ABLATION RESCALE FACTOR

Table 5: **NATS-Bench topology search space classification results on ConNAS with different rescale factors.** Test accuracy (mean ± std over 5 runs) is reported under hardware constraints, using on-device measurements of inference latency and energy usage on an NVIDIA Jetson TX2 Edge GPU. The constraints are chosen so that approximately 50% of architectures in the search space satisfy the constraints. The relative error compared to the optimal architecture satisfying the hardware constraints is reported between parentheses.

| Rescaling factor | Top-1 accuracy (%) | | | Runs satisfied |
|---|---|---|---|---|
| | **CIFAR-10** | **CIFAR-100** | **ImageNet16-120** | |
| $c_{\text{ts,latency}}$ : latency $\leq 5.31ms$ *(7814 architectures satisfied)* | | | | |
| 1.2 | $93.18_{\pm 0.09}$ *(-1.08)* | $70.12_{\pm 0.16}$ *(-3.04)* | $40.59_{\pm 1.20}$ *(-6.01)* | 5 / 5 |
| 1.5 | $93.20_{\pm 0.09}$ *(-1.06)* | $70.07_{\pm 0.13}$ *(-3.09)* | $40.16_{\pm 0.97}$ *(-6.44)* | 5 / 5 |
| 2.0 | $93.21_{\pm 0.05}$ *(-1.05)* | $70.19_{\pm 0.12}$ *(-2.97)* | $41.04_{\pm 1.18}$ *(-5.56)* | 5 / 5 |
| 5.0 | $93.25_{\pm 0.04}$ *(-1.02)* | $70.10_{\pm 0.10}$ *(-3.06)* | $40.18_{\pm 0.96}$ *(-6.42)* | 5 / 5 |
| 10.0 | $92.74_{\pm 0.67}$ *(-1.52)* | $68.83_{\pm 1.79}$ *(-4.33)* | $39.60_{\pm 1.48}$ *(-7.00)* | 5 / 5 |
| $c_{\text{ts,energy}}$ : energy usage $\leq 23.95mJ$ *(7814 architectures satisfied)* | | | | |
| 1.2 | $92.97_{\pm 0.54}$ *(-1.18)* | $69.51_{\pm 1.48}$ *(-2.43)* | $40.32_{\pm 1.56}$ *(-6.45)* | 5 / 5 |
| 1.5 | $93.20_{\pm 0.09}$ *(-0.95)* | $70.07_{\pm 0.13}$ *(-1.87)* | $40.16_{\pm 0.97}$ *(-6.61)* | 5 / 5 |
| 2.0 | $92.95_{\pm 0.53}$ *(-1.20)* | $69.55_{\pm 1.50}$ *(-2.39)* | $40.75_{\pm 1.66}$ *(-6.02)* | 5 / 5 |
| 5.0 | $93.17_{\pm 0.00}$ *(-0.98)* | $70.28_{\pm 0.00}$ *(-1.66)* | $41.90_{\pm 0.00}$ *(-4.87)* | 5 / 5 |
| 10.0 | $92.72_{\pm 0.65}$ *(-1.43)* | $68.87_{\pm 1.83}$ *(-3.07)* | $40.03_{\pm 1.81}$ *(-6.74)* | 5 / 5 |
| $c_{\text{ts,comb}}$ : latency $\leq 5.31ms \wedge$ energy usage $\leq 23.95mJ$ *(7586 architectures satisfied)* | | | | |
| 1.2 | $93.14_{\pm 0.05}$ *(-1.01)* | $70.21_{\pm 0.17}$ *(-1.73)* | $41.45_{\pm 1.01}$ *(-5.08)* | 5 / 5 |
| 1.5 | $92.95_{\pm 0.53}$ *(-1.20)* | $69.55_{\pm 1.50}$ *(-2.39)* | $40.75_{\pm 1.66}$ *(-5.78)* | 5 / 5 |
| 2.0 | $93.17_{\pm 0.00}$ *(-0.98)* | $70.28_{\pm 0.00}$ *(-1.66)* | $41.90_{\pm 0.00}$ *(-4.63)* | 5 / 5 |
| 5.0 | $92.89_{\pm 0.50}$ *(-1.26)* | $69.45_{\pm 1.45}$ *(-2.49)* | $40.28_{\pm 1.58}$ *(-6.25)* | 5 / 5 |
| 10.0 | $92.49_{\pm 0.66}$ *(-1.66)* | $68.19_{\pm 1.81}$ *(-3.75)* | $39.31_{\pm 1.58}$ *(-7.22)* | 5 / 5 |

Table 6: **NATS-Bench size search space classification results on ConNAS with different rescale factors.** Test accuracy (mean ± std over 5 runs) is reported under hardware constraints, including the number of parameters, number of FLOPs, and peak memory usage. The constraints are chosen so that about 50% of the architectures in each search space meet them. The relative error compared to the optimal architecture satisfying the hardware constraints is reported between parentheses.

| Rescaling factor | Top-1 accuracy (%) | | | Runs satisfied |
|---|---|---|---|---|
| | **CIFAR-10** | **CIFAR-100** | **ImageNet16-120** | |
| $c_{\text{ss,param}}$ : #parameters $\leq 261650$ *(16385 architectures satisfied)* | | | | |
| 1.2 | $91.74_{\pm 0.29\ (-0.79)}$ | $66.22_{\pm 0.47\ (-2.70)}$ | $39.77_{\pm 0.77\ (-2.27)}$ | 5 / 5 |
| 1.5 | $92.09_{\pm 0.24\ (-0.44)}$ | $65.92_{\pm 0.65\ (-3.00)}$ | $40.29_{\pm 1.10\ (-1.74)}$ | 5 / 5 |
| 2.0 | $91.87_{\pm 0.07\ (-0.66)}$ | $66.22_{\pm 0.60\ (-2.70)}$ | $39.48_{\pm 1.29\ (-2.55)}$ | 5 / 5 |
| 5.0 | $91.85_{\pm 0.19\ (-0.68)}$ | $65.97_{\pm 0.38\ (-2.95)}$ | $39.55_{\pm 0.53\ (-2.48)}$ | 5 / 5 |
| 10.0 | $91.67_{\pm 0.08\ (-0.86)}$ | $65.99_{\pm 0.36\ (-2.93)}$ | $39.31_{\pm 0.21\ (-2.73)}$ | 5 / 5 |
| $c_{\text{ss,mem}}$ : peak memory usage $\leq 655kB$ *(20480 architectures satisfied)* | | | | |
| 1.2 | $93.28_{\pm 0.20\ (-0.14)}$ | $69.70_{\pm 1.02\ (-1.16)}$ | $44.58_{\pm 0.43\ (-1.35)}$ | 5 / 5 |
| 1.5 | $93.26_{\pm 0.15\ (-0.16)}$ | $70.14_{\pm 0.32\ (-0.72)}$ | $44.67_{\pm 0.68\ (-1.26)}$ | 5 / 5 |
| 2.0 | $93.19_{\pm 0.17\ (-0.23)}$ | $69.54_{\pm 0.95\ (-1.32)}$ | $44.42_{\pm 0.38\ (-1.51)}$ | 5 / 5 |
| 5.0 | $93.26_{\pm 0.16\ (-0.16)}$ | $70.06_{\pm 0.15\ (-0.80)}$ | $44.65_{\pm 0.18\ (-1.29)}$ | 5 / 5 |
| 10.0 | $93.29_{\pm 0.11\ (-0.13)}$ | $70.01_{\pm 0.24\ (-0.85)}$ | $44.62_{\pm 0.22\ (-1.31)}$ | 5 / 5 |
| $c_{\text{ss,flops}}$ : #FLOPs $\leq 1196M$ *(16385 architectures satisfied)* | | | | |
| 1.2 | $91.71_{\pm 0.28\ (-1.09)}$ | $65.44_{\pm 0.80\ (-4.10)}$ | $38.33_{\pm 0.36\ (-5.13)}$ | 5 / 5 |
| 1.5 | $91.73_{\pm 0.27\ (-1.07)}$ | $65.80_{\pm 1.31\ (-3.74)}$ | $38.73_{\pm 0.53\ (-4.73)}$ | 5 / 5 |
| 2.0 | $91.47_{\pm 0.25\ (-1.33)}$ | $65.38_{\pm 0.58\ (-4.16)}$ | $38.70_{\pm 1.17\ (-4.77)}$ | 5 / 5 |
| 5.0 | $91.48_{\pm 0.32\ (-1.32)}$ | $66.03_{\pm 1.16\ (-3.51)}$ | $38.56_{\pm 1.45\ (-4.91)}$ | 5 / 5 |
| 10.0 | $91.88_{\pm 0.26\ (-0.92)}$ | $66.18_{\pm 1.20\ (-3.36)}$ | $38.83_{\pm 1.07\ (-4.63)}$ | 5 / 5 |
| $c_{\text{ss,comb}}$ : #parameters $\leq 261650 \wedge$ peak memory usage $\leq 655kB\wedge$ #FLOPs $\leq 1196M$ *(13956 architectures satisfied)* | | | | |
| 1.2 | $91.72_{\pm 0.35\ (-0.65)}$ | $65.95_{\pm 0.52\ (-2.97)}$ | $39.52_{\pm 0.46\ (-2.25)}$ | 5 / 5 |
| 1.5 | $92.00_{\pm 0.21\ (-0.37)}$ | $66.72_{\pm 0.71\ (-2.19)}$ | $39.54_{\pm 1.13\ (-2.23)}$ | 5 / 5 |
| 2.0 | $91.52_{\pm 0.61\ (-0.85)}$ | $65.70_{\pm 1.16\ (-3.22)}$ | $38.79_{\pm 1.73\ (-2.97)}$ | 5 / 5 |
| 5.0 | $91.73_{\pm 0.25\ (-0.64)}$ | $65.53_{\pm 0.29\ (-3.39)}$ | $38.78_{\pm 0.47\ (-2.99)}$ | 5 / 5 |
| 10.0 | $91.88_{\pm 0.11\ (-0.49)}$ | $65.85_{\pm 0.34\ (-3.07)}$ | $39.31_{\pm 1.04\ (-2.45)}$ | 5 / 5 |

## C    EXPERIMENTS WITH STRICTER HARDWARE CONSTRAINTS

Table 7: **NATS-Bench topology search space classification results under stricter hardware constraints.** Test accuracy (mean ± std over 5 runs) is reported under hardware constraints, using on-device measurements of inference latency and energy usage on an NVIDIA Jetson TX2 Edge GPU. Stricter hardware constraints were set so that approximately 25% and 10% of architectures in the search space satisfy the constraints. The relative error compared to the optimal architecture satisfying the hardware constraints is reported between parentheses.

| Method | Top-1 accuracy (%) | | | Runs satisfied |
|---|---|---|---|---|
| | **CIFAR-10** | **CIFAR-100** | **ImageNet16-120** | |
| $c_{\text{ts,latency25}}$ : latency $\leq 4.14ms$ *(3908 architectures satisfied)* | | | | |
| ConNAS | $92.08 \pm 0.07$ *(-1.97)* | $67.09 \pm 0.20$ *(-4.43)* | $39.09 \pm 0.72$ *(-5.88)* | 5 / 5 |
| **ProxylessNAS** | $\mathbf{92.29} \pm 0.00$ *(-1.76)* | $\mathbf{67.48} \pm 0.00$ *(-4.04)* | $\mathbf{39.40} \pm 0.00$ *(-5.57)* | 5 / 5 |
| FBNet | $93.22 \pm 0.09$ | $70.03 \pm 0.07$ | $39.73 \pm 0.05$ | 0 / 5 |
| TAS | $93.27 \pm 0.00$ | $70.06 \pm 0.00$ | $39.75 \pm 0.00$ | 0 / 5 |
| **TF-NAS** | $\mathbf{92.29} \pm 0.00$ *(-1.76)* | $\mathbf{67.48} \pm 0.00$ *(-4.04)* | $\mathbf{39.40} \pm 0.00$ *(-5.57)* | 5 / 5 |
| HDX | $10.00 \pm 0.00$ *(-84.05)* | $1.00 \pm 0.00$ *(-70.52)* | $0.83 \pm 0.00$ *(-44.13)* | 5 / 5 |
| $c_{\text{ts,latency10}}$ : latency $\leq 3.03ms$ *(1564 architectures satisfied)* | | | | |
| **ConNAS** | $\mathbf{92.05} \pm 0.07$ *(-0.73)* | $\mathbf{67.01} \pm 0.20$ *(-1.35)* | $\mathbf{38.83} \pm 0.72$ *(-1.61)* | 5 / 5 |
| ProxylessNAS | $92.29 \pm 0.00$ | $67.48 \pm 0.00$ | $39.40 \pm 0.00$ | 0 / 5 |
| FBNet | $93.22 \pm 0.09$ | $70.03 \pm 0.07$ | $39.73 \pm 0.05$ | 0 / 5 |
| TAS | $92.00 \pm 0.01$ | $67.13 \pm 0.05$ | $39.13 \pm 0.05$ | 0 / 5 |
| TF-NAS | $10.00 \pm 0.00$ *(-82.78)* | $1.00 \pm 0.00$ *(-67.36)* | $0.83 \pm 0.00$ *(-39.60)* | 5 / 5 |
| HDX | $10.00 \pm 0.00$ *(-82.78)* | $1.00 \pm 0.00$ *(-67.36)* | $0.83 \pm 0.00$ *(-39.60)* | 5 / 5 |
| $c_{\text{ts,energy25}}$ : energy usage $\leq 18.6mJ$ *(3908 architectures satisfied)* | | | | |
| ConNAS | $92.06 \pm 0.06$ *(-1.98)* | $67.09 \pm 0.20$ *(-4.43)* | $39.04 \pm 0.68$ *(-5.93)* | 5 / 5 |
| **ProxylessNAS** | $\mathbf{92.29} \pm 0.00$ *(-1.76)* | $\mathbf{67.48} \pm 0.00$ *(-4.04)* | $\mathbf{39.40} \pm 0.00$ *(-5.57)* | 5 / 5 |
| FBNet | $93.14 \pm 0.12$ | $69.97 \pm 0.08$ | $39.69 \pm 0.06$ | 0 / 5 |
| TAS | $93.01 \pm 0.57$ | $69.48 \pm 1.29$ | $39.62 \pm 0.30$ | 1 / 5 |
| **TF-NAS** | $\mathbf{92.29} \pm 0.00$ *(-1.76)* | $\mathbf{67.48} \pm 0.00$ *(-4.04)* | $\mathbf{39.40} \pm 0.00$ *(-5.57)* | 5 / 5 |
| HDX | $10.00 \pm 0.00$ *(-84.05)* | $1.00 \pm 0.00$ *(-70.52)* | $0.83 \pm 0.00$ *(-44.13)* | 5 / 5 |
| $c_{\text{ts,energy10}}$ : energy usage $\leq 13.3mJ$ *(1564 architectures satisfied)* | | | | |
| ConNAS | $92.05 \pm 0.07$ *(-0.73)* | $67.01 \pm 0.20$ *(-1.35)* | $38.83 \pm 0.72$ *(-1.61)* | 5 / 5 |
| **ProxylessNAS** | $\mathbf{92.29} \pm 0.00$ *(-0.49)* | $\mathbf{67.48} \pm 0.00$ *(-0.88)* | $\mathbf{39.40} \pm 0.00$ *(-1.03)* | 5 / 5 |
| FBNet | $93.14 \pm 0.12$ | $69.97 \pm 0.08$ | $39.69 \pm 0.06$ | 0 / 5 |
| TAS | $92.00 \pm 0.01$ | $67.11 \pm 0.05$ | $39.15 \pm 0.05$ | 3 / 5 |
| TF-NAS | $10.00 \pm 0.00$ *(-82.78)* | $1.00 \pm 0.00$ *(-67.36)* | $0.83 \pm 0.00$ *(-39.60)* | 5 / 5 |
| HDX | $10.00 \pm 0.00$ *(-82.78)* | $1.00 \pm 0.00$ *(-67.36)* | $0.83 \pm 0.00$ *(-39.60)* | 5 / 5 |
| $c_{\text{ts,comb25}}$ : latency $\leq 4.14ms \wedge$ energy usage $\leq 18.6mJ$ *(3785 architectures satisfied)* | | | | |
| ConNAS | $92.10 \pm 0.06$ *(-1.95)* | $67.16 \pm 0.16$ *(-4.36)* | $39.35 \pm 0.59$ *(-5.61)* | 5 / 5 |
| **ProxylessNAS** | $\mathbf{92.29} \pm 0.00$ *(-1.76)* | $\mathbf{67.48} \pm 0.00$ *(-4.04)* | $\mathbf{39.40} \pm 0.00$ *(-5.57)* | 5 / 5 |
| FBNet | $92.01 \pm 0.00$ *(-2.04)* | $67.07 \pm 0.00$ *(-4.45)* | $39.19 \pm 0.00$ *(-5.78)* | 5 / 5 |
| TAS | $93.27 \pm 0.00$ | $70.06 \pm 0.00$ | $39.75 \pm 0.00$ | 0 / 5 |
| **TF-NAS** | $\mathbf{92.29} \pm 0.00$ *(-1.76)* | $\mathbf{67.48} \pm 0.00$ *(-4.04)* | $\mathbf{39.40} \pm 0.00$ *(-5.57)* | 5 / 5 |
| HDX | $10.00 \pm 0.00$ *(-84.05)* | $1.00 \pm 0.00$ *(-70.52)* | $0.83 \pm 0.00$ *(-44.13)* | 5 / 5 |
| $c_{\text{ts,comb10}}$ : latency $\leq 3.03ms \wedge$ energy usage $\leq 13.3mJ$ *(1526 architectures satisfied)* | | | | |
| **ConNAS** | $\mathbf{92.08} \pm 0.06$ *(-0.70)* | $\mathbf{67.20} \pm 0.21$ *(-1.16)* | $\mathbf{39.10} \pm 0.70$ *(-1.33)* | 5 / 5 |
| ProxylessNAS | $92.29 \pm 0.00$ | $67.48 \pm 0.00$ | $39.40 \pm 0.00$ | 0 / 5 |
| FBNet | $92.01 \pm 0.00$ | $67.07 \pm 0.00$ | $39.19 \pm 0.00$ | 0 / 5 |
| TAS | $91.99 \pm 0.01$ | $67.15 \pm 0.04$ | $39.11 \pm 0.04$ | 0 / 5 |
| TF-NAS | $10.00 \pm 0.00$ *(-82.78)* | $1.00 \pm 0.00$ *(-67.36)* | $0.83 \pm 0.00$ *(-39.60)* | 5 / 5 |
| HDX | $10.00 \pm 0.00$ *(-82.78)* | $1.00 \pm 0.00$ *(-67.36)* | $0.83 \pm 0.00$ *(-39.60)* | 5 / 5 |

Table 8: **NATS-Bench size search space classification results under stricter hardware constraints.** Test accuracy (mean ± std over 5 runs) is reported under hardware constraints, including the number of parameters, number of FLOPs, and peak memory usage. Stricter hardware constraints were set so that approximately 25% and 10% of architectures in the search space satisfy the constraints. The relative error compared to the optimal architecture satisfying the hardware constraints is reported between parentheses.

| Method | Top-1 accuracy (%) | | | Runs satisfied |
|---|---|---|---|---|
| | **CIFAR-10** | **CIFAR-100** | **ImageNet16-120** | |
| $c_{\mathrm{ss,param25}}$ : #parameters $\leq 187210$ *(8193 architectures satisfied)* | | | | |
| **ConNAS** | **91.33** ± 0.18 *(-0.50)* | **64.03** ± 0.66 *(-2.55)* | **38.25** ± 0.29 *(-1.01)* | 5 / 5 |
| ProxylessNAS | 85.87 ± 1.16 *(-5.96)* | 51.11 ± 2.35 *(-15.47)* | 25.27 ± 2.13 *(-13.99)* | 5 / 5 |
| FBNet | 93.22 ± 0.20 | 69.43 ± 0.22 | 45.68 ± 0.37 | 0 / 5 |
| TAS | 91.55 ± 0.31 | 64.09 ± 0.61 | 39.01 ± 1.39 | 0 / 5 |
| TF-NAS | 89.43 ± 0.64 *(-2.40)* | 58.75 ± 2.84 *(-7.83)* | 31.99 ± 1.56 *(-7.28)* | 5 / 5 |
| HDX | 79.79 ± 0.00 *(-12.04)* | 32.30 ± 0.00 *(-34.28)* | 15.53 ± 0.00 *(-23.73)* | 5 / 5 |
| $c_{\mathrm{ss,param10}}$ : #parameters $\leq 78906$ *(3278 architectures satisfied)* | | | | |
| **ConNAS** | **90.04** ± 0.53 *(-1.06)* | **60.86** ± 2.01 *(-3.34)* | **34.09** ± 0.83 *(-2.71)* | 5 / 5 |
| ProxylessNAS | 85.87 ± 1.16 *(-5.23)* | 51.11 ± 2.35 *(-13.09)* | 25.27 ± 2.13 *(-11.53)* | 5 / 5 |
| FBNet | 93.22 ± 0.20 | 69.43 ± 0.22 | 45.68 ± 0.37 | 0 / 5 |
| TAS | 91.08 ± 0.40 | 63.75 ± 0.09 | 37.04 ± 0.83 | 0 / 5 |
| TF-NAS | 85.62 ± 0.00 *(-5.48)* | 47.74 ± 0.00 *(-16.46)* | 25.43 ± 0.00 *(-11.37)* | 5 / 5 |
| HDX | 79.79 ± 0.00 *(-11.31)* | 32.30 ± 0.00 *(-31.90)* | 15.53 ± 0.00 *(-21.27)* | 5 / 5 |
| $c_{\mathrm{ss,memory25}}$ : peak memory usage $\leq 393kB$ *(12288 architectures satisfied)* | | | | |
| **ConNAS** | **92.91** ± 0.08 *(-0.21)* | 69.28 ± 0.24 *(-1.20)* | 43.65 ± 0.10 *(-1.05)* | 5 / 5 |
| ProxylessNAS | 91.33 ± 0.19 *(-1.79)* | 68.22 ± 0.28 *(-2.26)* | 38.49 ± 0.03 *(-6.21)* | 5 / 5 |
| FBNet | 93.16 ± 0.29 | 69.69 ± 0.05 | 45.53 ± 0.87 | 0 / 5 |
| **TAS** | 92.82 ± 0.28 *(-0.30)* | 69.40 ± 0.46 *(-1.08)* | **43.69** ± 0.49 *(-1.01)* | 5 / 5 |
| **TF-NAS** | 92.35 ± 0.09 *(-0.77)* | **69.52** ± 0.13 *(-0.96)* | 43.09 ± 0.61 *(-1.61)* | 5 / 5 |
| HDX | 79.79 ± 0.00 *(-13.33)* | 32.30 ± 0.00 *(-38.18)* | 15.53 ± 0.00 *(-29.17)* | 5 / 5 |
| $c_{\mathrm{ss,memory10}}$ : peak memory usage $\leq 229kB$ *(3584 architectures satisfied)* | | | | |
| **ConNAS** | **92.30** ± 0.17 *(-0.50)* | **68.80** ± 0.32 *(-0.50)* | **40.80** ± 0.65 *(-1.30)* | 5 / 5 |
| ProxylessNAS | 91.33 ± 0.19 *(-1.47)* | 68.22 ± 0.28 *(-1.08)* | 38.49 ± 0.03 *(-3.61)* | 5 / 5 |
| FBNet | 93.16 ± 0.29 | 69.69 ± 0.05 | 45.53 ± 0.87 | 0 / 5 |
| TAS | 92.31 ± 0.16 | 68.67 ± 0.46 | 41.98 ± 0.91 | 0 / 5 |
| TF-NAS | 91.33 ± 0.19 *(-1.47)* | 68.22 ± 0.28 *(-1.08)* | 38.49 ± 0.03 *(-3.61)* | 5 / 5 |
| HDX | 79.79 ± 0.00 *(-13.01)* | 32.30 ± 0.00 *(-37.00)* | 15.53 ± 0.00 *(-26.57)* | 5 / 5 |
| $c_{\mathrm{ss,flops25}}$ : #FLOPs $\leq 514M$ *(8193 architectures satisfied)* | | | | |
| **ConNAS** | **90.50** ± 0.23 *(-1.48)* | 63.30 ± 0.39 *(-5.22)* | **34.87** ± 0.92 *(-6.03)* | 5 / 5 |
| ProxylessNAS | 89.55 ± 0.22 *(-2.43)* | **63.51** ± 0.67 *(-5.01)* | 32.43 ± 0.84 *(-8.47)* | 5 / 5 |
| FBNet | 93.02 ± 0.09 | 69.28 ± 0.36 | 45.04 ± 0.60 | 0 / 5 |
| TAS | 91.80 ± 0.30 | 68.04 ± 0.60 | 40.45 ± 0.37 | 0 / 5 |
| TF-NAS | 89.25 ± 0.12 *(-2.73)* | 63.41 ± 0.07 *(-5.11)* | 33.31 ± 1.35 *(-7.59)* | 5 / 5 |
| HDX | 79.79 ± 0.00 *(-12.19)* | 32.30 ± 0.00 *(-36.22)* | 15.53 ± 0.00 *(-25.37)* | 5 / 5 |
| $c_{\mathrm{ss,flops10}}$ : #FLOPs $\leq 223M$ *(3278 architectures satisfied)* | | | | |
| **ConNAS** | **89.85** ± 0.52 *(-1.31)* | 61.35 ± 1.46 *(-5.99)* | 33.57 ± 0.89 *(-4.89)* | 5 / 5 |
| **ProxylessNAS** | 89.55 ± 0.22 *(-1.61)* | **63.51** ± 0.67 *(-3.83)* | 32.43 ± 0.84 *(-6.03)* | 5 / 5 |
| FBNet | 93.02 ± 0.09 | 69.28 ± 0.36 | 45.04 ± 0.60 | 0 / 5 |
| TAS | 91.39 ± 0.49 | 66.80 ± 0.41 | 38.04 ± 0.70 | 0 / 5 |
| **TF-NAS** | 89.29 ± 0.09 *(-1.87)* | 63.38 ± 0.05 *(-3.96)* | **33.81** ± 1.10 *(-4.66)* | 5 / 5 |
| HDX | 79.79 ± 0.00 *(-11.37)* | 32.30 ± 0.00 *(-35.04)* | 15.53 ± 0.00 *(-22.93)* | 5 / 5 |

| $c_{\text{ss,comb25}}$ : #parameters $\leq 187210 \wedge$ peak memory usage $\leq 393kB \wedge$ | | | |
|---|---|---|---|
| #FLOPs $\leq 514M$ *(4446 architectures satisfied)* | | | |
| **ConNAS** | **90.87** $_{\pm 0.18}$ *(-0.75)* | **64.61** $_{\pm 0.83}$ *(-1.97)* | **36.60** $_{\pm 1.21}$ *(-2.67)* | 5 / 5 |
| ProxylessNAS | 84.12 $_{\pm 0.28}$ *(-7.50)* | 49.12 $_{\pm 1.07}$ *(-17.46)* | 22.64 $_{\pm 0.31}$ *(-16.63)* | 5 / 5 |
| FBNet | 92.59 $_{\pm 0.32}$ | 67.92 $_{\pm 0.64}$ | 43.09 $_{\pm 1.12}$ | 0 / 5 |
| TAS | 91.31 $_{\pm 0.22}$ | 65.00 $_{\pm 0.95}$ | 37.95 $_{\pm 0.74}$ | 0 / 5 |
| TF-NAS | 87.42 $_{\pm 0.62}$ *(-4.20)* | 57.65 $_{\pm 0.85}$ *(-8.93)* | 27.80 $_{\pm 1.33}$ *(-11.47)* | 5 / 5 |
| HDX | 79.79 $_{\pm 0.00}$ *(-11.83)* | 32.30 $_{\pm 0.00}$ *(-34.28)* | 15.53 $_{\pm 0.00}$ *(-23.73)* | 5 / 5 |
| $c_{\text{ss,comb10}}$ : #parameters $\leq 78906 \wedge$ peak memory usage $\leq 229kB \wedge$ | | | |
| #FLOPs $\leq 223M$ *977 architectures satisfied* | | | |
| **ConNAS** | **89.66** $_{\pm 0.46}$ *(-0.66)* | **62.06** $_{\pm 0.73}$ *(-1.02)* | **32.57** $_{\pm 1.35}$ *(-1.73)* | 5 / 5 |
| ProxylessNAS | 84.12 $_{\pm 0.28}$ *(-6.20)* | 49.12 $_{\pm 1.07}$ *(-13.96)* | 22.64 $_{\pm 0.31}$ *(-11.66)* | 5 / 5 |
| FBNet | 92.59 $_{\pm 0.32}$ | 67.92 $_{\pm 0.64}$ | 43.09 $_{\pm 1.12}$ | 0 / 5 |
| TAS | 91.53 $_{\pm 0.14}$ | 64.32 $_{\pm 0.36}$ | 37.10 $_{\pm 0.18}$ | 0 / 5 |
| TF-NAS | 86.46 $_{\pm 0.11}$ *(-3.86)* | 56.76 $_{\pm 0.27}$ *(-6.32)* | 25.81 $_{\pm 0.51}$ *(-8.49)* | 5 / 5 |
| HDX | 79.79 $_{\pm 0.00}$ *(-10.53)* | 32.30 $_{\pm 0.00}$ *(-30.78)* | 15.53 $_{\pm 0.00}$ *(-18.77)* | 5 / 5 |

# D ABLATION WEIGHTING FACTOR BASELINE METHODS

Table 9: **Comparison of classification results on NATS-Bench topology search space across baseline methods and weighting factors.** Test accuracy (mean ± std over 5 runs) is reported under hardware constraints, using on-device measurements of inference latency and energy usage on an NVIDIA Jetson TX2 Edge GPU. The relative error compared to the optimal architecture satisfying the hardware constraints is reported between parentheses. Best results for each method, constraint, and dataset are highlighted in bold. Performance of experiments that do not satisfy the hardware constraints are reported in gray.

| Method | $\lambda_{\text{hardware}}$ | Top-1 accuracy (%) | | | Runs satisfied |
|---|---|---|---|---|---|
| | | CIFAR-10 | CIFAR-100 | ImageNet16-120 | |
| $c_{\text{ts,latency}}$ : latency $\leq 5.31ms$ *(7814 architectures satisfied)* | | | | | |
| ProxylessNAS | 0.5 | $92.06_{\pm 0.13}$ *(-2.20)* | $67.15_{\pm 0.18}$ *(-6.01)* | $39.23_{\pm 0.09}$ *(-7.37)* | 5 / 5 |
| **ProxylessNAS** | **1.0** | $\mathbf{92.29}_{\pm 0.00}$ *(-1.97)* | $\mathbf{67.48}_{\pm 0.00}$ *(-5.68)* | $\mathbf{39.40}_{\pm 0.00}$ *(-7.20)* | 5 / 5 |
| ProxylessNAS | 1.5 | $42.92_{\pm 45.07}$ *(-51.34)* | $27.59_{\pm 36.41}$ *(-45.57)* | $16.26_{\pm 21.12}$ *(-30.34)* | 5 / 5 |
| FBNet | 0.5 | $93.10_{\pm 0.09}$ *(-1.16)* | $69.94_{\pm 0.07}$ *(-3.22)* | $39.67_{\pm 0.05}$ *(-6.93)* | 5 / 5 |
| FBNet | 0.6 | $93.18_{\pm 0.12}$ *(-1.08)* | $70.00_{\pm 0.08}$ *(-3.16)* | $39.71_{\pm 0.06}$ *(-6.89)* | 5 / 5 |
| **FBNet** | **0.7** | $\mathbf{93.27}_{\pm 0.00}$ *(-1.00)* | $\mathbf{70.06}_{\pm 0.00}$ *(-3.10)* | $\mathbf{39.75}_{\pm 0.00}$ *(-6.85)* | 5 / 5 |
| TAS | 0.5 | $93.67_{\pm 0.00}$ | $70.91_{\pm 0.00}$ | $41.02_{\pm 0.00}$ | 0 / 5 |
| TAS | 1.0 | $93.67_{\pm 0.00}$ | $70.91_{\pm 0.00}$ | $41.02_{\pm 0.00}$ | 0 / 5 |
| TAS | 1.5 | $93.67_{\pm 0.00}$ | $70.91_{\pm 0.00}$ | $41.02_{\pm 0.00}$ | 0 / 5 |
| **TF-NAS** | **0.5** | $\mathbf{93.27}_{\pm 0.00}$ *(-1.00)* | $\mathbf{70.06}_{\pm 0.00}$ *(-3.10)* | $\mathbf{39.75}_{\pm 0.00}$ *(-6.85)* | 5 / 5 |
| **TF-NAS** | **1.0** | $\mathbf{93.27}_{\pm 0.00}$ *(-1.00)* | $\mathbf{70.06}_{\pm 0.00}$ *(-3.10)* | $\mathbf{39.75}_{\pm 0.00}$ *(-6.85)* | 5 / 5 |
| TF-NAS | 1.5 | $93.01_{\pm 0.57}$ *(-1.25)* | $69.48_{\pm 1.29}$ *(-3.68)* | $39.62_{\pm 0.30}$ *(-6.98)* | 5 / 5 |
| HDX | 0.5 | $42.86_{\pm 44.99}$ *(-51.40)* | $27.51_{\pm 36.30}$ *(-45.65)* | $16.22_{\pm 21.07}$ *(-30.38)* | 5 / 5 |
| **HDX** | **1.0** | $\mathbf{75.83}_{\pm 36.80}$ *(-18.43)* | $\mathbf{54.18}_{\pm 29.73}$ *(-18.98)* | $\mathbf{31.69}_{\pm 17.25}$ *(-14.91)* | 5 / 5 |
| HDX | 1.5 | $59.37_{\pm 45.07}$ *(-34.89)* | $40.89_{\pm 36.41}$ *(-32.27)* | $23.97_{\pm 21.12}$ *(-22.63)* | 5 / 5 |
| $c_{\text{ts,energy}}$ : energy usage $\leq 23.95mJ$ *(7814 architectures satisfied)* | | | | | |
| **ProxylessNAS** | **0.5** | $\mathbf{92.29}_{\pm 0.00}$ *(-1.86)* | $\mathbf{67.48}_{\pm 0.00}$ *(-4.46)* | $\mathbf{39.40}_{\pm 0.00}$ *(-7.37)* | 5 / 5 |
| **ProxylessNAS** | **1.0** | $\mathbf{92.29}_{\pm 0.00}$ *(-1.86)* | $\mathbf{67.48}_{\pm 0.00}$ *(-4.46)* | $\mathbf{39.40}_{\pm 0.00}$ *(-7.37)* | 5 / 5 |
| ProxylessNAS | 1.5 | $26.46_{\pm 36.80}$ *(-67.69)* | $14.30_{\pm 29.73}$ *(-57.64)* | $8.55_{\pm 17.25}$ *(-38.22)* | 5 / 5 |
| FBNet | 0.5 | $93.05_{\pm 0.00}$ *(-1.10)* | $69.91_{\pm 0.00}$ *(-2.03)* | $39.64_{\pm 0.00}$ *(-7.12)* | 5 / 5 |
| FBNet | 0.6 | $93.05_{\pm 0.00}$ *(-1.10)* | $69.91_{\pm 0.00}$ *(-2.03)* | $39.64_{\pm 0.00}$ *(-7.12)* | 5 / 5 |
| **FBNet** | **0.7** | $\mathbf{93.22}_{\pm 0.09}$ *(-0.93)* | $\mathbf{70.03}_{\pm 0.07}$ *(-1.91)* | $\mathbf{39.73}_{\pm 0.05}$ *(-7.04)* | 5 / 5 |
| TAS | 0.5 | $93.67_{\pm 0.00}$ | $70.91_{\pm 0.00}$ | $41.02_{\pm 0.00}$ | 0 / 5 |
| TAS | 1.0 | $93.67_{\pm 0.00}$ | $70.91_{\pm 0.00}$ | $41.02_{\pm 0.00}$ | 0 / 5 |
| TAS | 1.5 | $93.67_{\pm 0.00}$ | $70.91_{\pm 0.00}$ | $41.02_{\pm 0.00}$ | 0 / 5 |
| **TF-NAS** | **0.5** | $\mathbf{93.27}_{\pm 0.00}$ *(-0.89)* | $\mathbf{70.06}_{\pm 0.00}$ *(-1.88)* | $\mathbf{39.75}_{\pm 0.00}$ *(-7.02)* | 5 / 5 |
| TF-NAS | 1.0 | $93.01_{\pm 0.57}$ *(-1.14)* | $69.48_{\pm 1.29}$ *(-2.46)* | $39.62_{\pm 0.30}$ *(-7.15)* | 5 / 5 |
| TF-NAS | 1.5 | $91.99_{\pm 0.01}$ *(-2.16)* | $67.15_{\pm 0.04}$ *(-4.79)* | $39.11_{\pm 0.04}$ *(-7.66)* | 5 / 5 |
| HDX | 0.5 | $42.92_{\pm 45.07}$ *(-51.23)* | $27.59_{\pm 36.41}$ *(-44.35)* | $16.26_{\pm 21.12}$ *(-30.51)* | 5 / 5 |
| HDX | 1.0 | $92.29_{\pm 0.00}$ *(-1.86)* | $67.48_{\pm 0.00}$ *(-4.46)* | $39.40_{\pm 0.00}$ *(-7.37)* | 5 / 5 |
| HDX | 1.5 | $10.00_{\pm 0.00}$ *(-84.15)* | $1.00_{\pm 0.00}$ *(-70.94)* | $0.83_{\pm 0.00}$ *(-45.93)* | 5 / 5 |
| $c_{\text{ts,comb}}$ : latency $\leq 5.31ms \land$ energy usage $\leq 23.95mJ$ *(7586 architectures satisfied)* | | | | | |
| **ProxylessNAS** | **0.5** | $\mathbf{92.29}_{\pm 0.00}$ *(-1.86)* | $\mathbf{67.48}_{\pm 0.00}$ *(-4.46)* | $\mathbf{39.40}_{\pm 0.00}$ *(-7.13)* | 5 / 5 |
| **ProxylessNAS** | **1.0** | $\mathbf{92.29}_{\pm 0.00}$ *(-1.86)* | $\mathbf{67.48}_{\pm 0.00}$ *(-4.46)* | $\mathbf{39.40}_{\pm 0.00}$ *(-7.13)* | 5 / 5 |
| ProxylessNAS | 1.5 | $42.92_{\pm 45.07}$ *(-51.23)* | $27.59_{\pm 36.41}$ *(-44.35)* | $16.26_{\pm 21.12}$ *(-30.27)* | 5 / 5 |
| **FBNet** | **0.7** | $\mathbf{92.01}_{\pm 0.00}$ *(-2.14)* | $\mathbf{67.07}_{\pm 0.00}$ *(-4.87)* | $\mathbf{39.19}_{\pm 0.00}$ *(-7.34)* | 5 / 5 |
| **FBNet** | **0.6** | $\mathbf{92.01}_{\pm 0.00}$ *(-2.14)* | $\mathbf{67.07}_{\pm 0.00}$ *(-4.87)* | $\mathbf{39.19}_{\pm 0.00}$ *(-7.34)* | 5 / 5 |
| **FBNet** | **0.7** | $\mathbf{92.01}_{\pm 0.00}$ *(-2.14)* | $\mathbf{67.07}_{\pm 0.00}$ *(-4.87)* | $\mathbf{39.19}_{\pm 0.00}$ *(-7.34)* | 5 / 5 |
| TAS | 0.5 | $93.67_{\pm 0.00}$ | $70.91_{\pm 0.00}$ | $41.02_{\pm 0.00}$ | 0 / 5 |
| TAS | 1.0 | $93.67_{\pm 0.00}$ | $70.91_{\pm 0.00}$ | $41.02_{\pm 0.00}$ | 0 / 5 |
| TAS | 1.5 | $93.67_{\pm 0.00}$ | $70.91_{\pm 0.00}$ | $41.02_{\pm 0.00}$ | 0 / 5 |
| **TF-NAS** | **0.5** | $\mathbf{93.27}_{\pm 0.00}$ *(-0.89)* | $\mathbf{70.06}_{\pm 0.00}$ *(-1.88)* | $\mathbf{39.75}_{\pm 0.00}$ *(-6.78)* | 5 / 5 |
| **TF-NAS** | **1.0** | $\mathbf{93.27}_{\pm 0.00}$ *(-0.89)* | $\mathbf{70.06}_{\pm 0.00}$ *(-1.88)* | $\mathbf{39.75}_{\pm 0.00}$ *(-6.78)* | 5 / 5 |
| TF-NAS | 1.5 | $92.76_{\pm 0.69}$ *(-1.39)* | $68.89_{\pm 1.61}$ *(-3.05)* | $39.51_{\pm 0.34}$ *(-7.02)* | 5 / 5 |
| HDX | 0.5 | $92.23_{\pm 0.13}$ | $67.40_{\pm 0.18}$ | $39.36_{\pm 0.09}$ | 1 / 5 |
| **HDX** | **1.0** | $\mathbf{92.29}_{\pm 0.00}$ *(-1.86)* | $\mathbf{67.48}_{\pm 0.00}$ *(-4.46)* | $\mathbf{39.40}_{\pm 0.00}$ *(-7.13)* | 5 / 5 |
| HDX | 1.5 | $59.37_{\pm 45.07}$ *(-34.78)* | $40.89_{\pm 36.41}$ *(-31.05)* | $23.97_{\pm 21.12}$ *(-22.56)* | 5 / 5 |

Table 10: **Comparison of classification results on NATS-Bench size search space across baseline methods and weighting factors.** Test accuracy (mean ± std over 5 runs) is reported under hardware constraints, including the number of parameters, number of FLOPs, and peak memory usage. The relative error compared to the optimal architecture satisfying the hardware constraints is reported between parentheses. Best results for each method, constraint, and dataset are highlighted in bold. Performance of experiments that do not satisfy the hardware constraints are reported in gray.

| Method | $\lambda_{\text{hardware}}$ | Top-1 accuracy (%) | | | Runs satisfied |
|---|---|---|---|---|---|
| | | **CIFAR-10** | **CIFAR-100** | **ImageNet16-120** | |
| $c_{\text{ss,param}}$ : #parameters $\leq 261650$ *(16385 architectures satisfied)* | | | | | |
| **ProxylessNAS** | **0.5** | **90.87** ± 0.02 *(-1.66)* | **61.31** ± 0.61 *(-7.61)* | **35.34** ± 0.31 *(-6.69)* | 5 / 5 |
| ProxylessNAS | 1.0 | 85.87 ± 1.16 *(-6.66)* | 51.11 ± 2.35 *(-17.81)* | 25.27 ± 2.13 *(-16.76)* | 5 / 5 |
| ProxylessNAS | 1.5 | 83.47 ± 0.74 *(-9.06)* | 45.36 ± 2.95 *(-23.56)* | 21.59 ± 1.01 *(-20.45)* | 5 / 5 |
| FBNet | 0.5 | 93.12 ± 0.17 | 69.36 ± 0.44 | 45.43 ± 0.48 | 0 / 5 |
| FBNet | 0.6 | 93.22 ± 0.20 | 69.43 ± 0.22 | 45.68 ± 0.37 | 0 / 5 |
| FBNet | 0.7 | 92.91 ± 0.26 | 69.36 ± 0.24 | 45.40 ± 0.41 | 0 / 5 |
| TAS | 0.5 | 92.26 ± 0.01 | 67.41 ± 0.38 | 42.10 ± 0.15 | 0 / 5 |
| TAS | 1.0 | 92.25 ± 0.02 | 67.34 ± 0.54 | 41.89 ± 0.33 | 0 / 5 |
| TAS | 1.5 | 92.25 ± 0.02 | 67.34 ± 0.54 | 41.89 ± 0.33 | 0 / 5 |
| TF-NAS | 0.5 | 91.30 ± 0.19 *(-1.23)* | **64.88** ± 0.28 *(-4.04)* | **40.43** ± 0.24 *(-1.61)* | 5 / 5 |
| **TF-NAS** | **1.0** | **91.78** ± 0.00 *(-0.75)* | 63.64 ± 0.00 *(-5.28)* | 38.00 ± 0.00 *(-4.03)* | 5 / 5 |
| TF-NAS | 1.5 | 91.56 ± 0.30 *(-0.97)* | 64.05 ± 0.56 *(-4.87)* | 38.17 ± 0.24 *(-3.86)* | 5 / 5 |
| **HDX** | **0.5** | **88.00** ± 0.91 *(-4.53)* | **54.74** ± 1.31 *(-14.18)* | **29.33** ± 3.50 *(-12.71)* | 5 / 5 |
| HDX | 1.0 | 86.10 ± 1.28 *(-6.43)* | 51.71 ± 2.88 *(-17.21)* | 25.36 ± 2.42 *(-16.67)* | 5 / 5 |
| HDX | 1.5 | 83.37 ± 0.88 *(-9.16)* | 45.20 ± 3.12 *(-23.72)* | 21.54 ± 1.51 *(-20.49)* | 5 / 5 |
| $c_{\text{ss,mem}}$ : peak memory usage $\leq 655kB$ *(20480 architectures satisfied)* | | | | | |
| ProxylessNAS | 0.5 | **91.61** ± 0.09 *(-1.81)* | 67.90 ± 0.18 *(-2.96)* | 38.30 ± 0.18 *(-7.63)* | 5 / 5 |
| **ProxylessNAS** | **1.0** | 91.33 ± 0.19 *(-2.09)* | **68.22** ± 0.28 *(-2.64)* | **38.49** ± 0.03 *(-7.45)* | 5 / 5 |
| **ProxylessNAS** | **1.5** | 91.33 ± 0.19 *(-2.09)* | **68.22** ± 0.28 *(-2.64)* | **38.49** ± 0.03 *(-7.45)* | 5 / 5 |
| FBNet | 0.5 | 93.24 ± 0.17 | 70.03 ± 0.38 | 45.43 ± 0.55 | 0 / 5 |
| FBNet | 0.6 | 93.16 ± 0.29 | 69.69 ± 0.05 | 45.53 ± 0.87 | 1 / 5 |
| FBNet | 0.7 | 93.23 ± 0.16 | 69.94 ± 0.34 | 45.71 ± 0.45 | 1 / 5 |
| TAS | 0.5 | 93.08 ± 0.20 *(-0.34)* | 70.23 ± 0.16 *(-0.63)* | **45.14** ± 0.31 *(-0.79)* | 5 / 5 |
| TAS | 1.0 | 92.93 ± 0.19 *(-0.49)* | 70.10 ± 0.32 *(-0.76)* | 44.98 ± 0.59 *(-0.95)* | 5 / 5 |
| **TAS** | **1.5** | **93.13** ± 0.19 *(-0.29)* | **70.30** ± 0.19 *(-0.56)* | 45.02 ± 0.32 *(-0.91)* | 5 / 5 |
| TF-NAS | 0.5 | 92.92 ± 0.31 *(-0.50)* | 69.94 ± 0.31 *(-0.92)* | 44.73 ± 0.67 *(-1.21)* | 5 / 5 |
| **TF-NAS** | **1.0** | 92.99 ± 0.10 *(-0.43)* | **70.21** ± 0.16 *(-0.65)* | **45.25** ± 0.27 *(-0.69)* | 5 / 5 |
| TF-NAS | 1.5 | **93.02** ± 0.28 *(-0.40)* | 70.13 ± 0.34 *(-0.73)* | 45.02 ± 0.61 *(-0.91)* | 5 / 5 |
| HDX | 0.5 | 89.43 ± 5.40 *(-3.99)* | 60.85 ± 15.96 *(-10.01)* | 34.23 ± 10.49 *(-11.70)* | 5 / 5 |
| HDX | 1.0 | **91.59** ± 0.19 *(-1.83)* | 68.03 ± 0.29 *(-2.83)* | 38.41 ± 0.07 *(-7.53)* | 5 / 5 |
| **HDX** | **1.5** | 91.41 ± 0.24 *(-2.01)* | **68.09** ± 0.34 *(-2.77)* | **38.47** ± 0.04 *(-7.46)* | 5 / 5 |
| $c_{\text{ss,flops}}$ : #FLOPs $\leq 1196M$ *(16385 architectures satisfied)* | | | | | |
| **ProxylessNAS** | **0.5** | **91.36** ± 0.14 *(-1.44)* | **66.93** ± 0.65 *(-2.61)* | **38.55** ± 0.52 *(-4.92)* | 5 / 5 |
| ProxylessNAS | 1.0 | 89.55 ± 0.22 *(-3.25)* | 63.51 ± 0.67 *(-6.03)* | 32.43 ± 0.84 *(-11.03)* | 5 / 5 |
| ProxylessNAS | 1.5 | 87.92 ± 0.47 *(-4.88)* | 60.76 ± 1.01 *(-8.78)* | 29.25 ± 0.64 *(-14.22)* | 5 / 5 |
| FBNet | 0.5 | 92.97 ± 0.06 | 69.29 ± 0.22 | 45.47 ± 0.25 | 0 / 5 |
| FBNet | 0.6 | 93.02 ± 0.09 | 69.28 ± 0.36 | 45.04 ± 0.60 | 0 / 5 |
| FBNet | 0.7 | 93.02 ± 0.06 | 69.44 ± 0.49 | 45.07 ± 0.46 | 0 / 5 |
| TAS | 0.5 | 92.46 ± 0.34 | 69.15 ± 0.45 | 43.31 ± 0.60 | 0 / 5 |
| TAS | 1.0 | 92.15 ± 0.19 | 68.66 ± 0.52 | 42.11 ± 0.82 | 3 / 5 |
| **TAS** | **1.5** | **92.13** ± 0.16 *(-0.67)* | **68.60** ± 0.55 *(-0.94)* | **41.93** ± 0.88 *(-1.53)* | 5 / 5 |
| **TF-NAS** | **0.5** | **91.21** ± 0.39 *(-1.59)* | **66.75** ± 0.74 *(-2.79)* | 37.50 ± 0.89 *(-5.97)* | 5 / 5 |
| TF-NAS | 1.0 | 91.10 ± 0.25 *(-1.70)* | 66.56 ± 1.12 *(-2.98)* | 37.41 ± 0.94 *(-6.06)* | 5 / 5 |
| TF-NAS | 1.5 | 90.98 ± 0.32 *(-1.82)* | 66.43 ± 0.52 *(-3.11)* | **37.63** ± 1.13 *(-5.84)* | 5 / 5 |
| HDX | 0.5 | 83.99 ± 0.00 *(-8.81)* | 48.64 ± 0.00 *(-20.90)* | 22.50 ± 0.00 *(-20.97)* | 5 / 5 |
| **HDX** | **1.0** | **84.43** ± 0.99 *(-8.37)* | **49.02** ± 0.85 *(-20.52)* | **23.17** ± 1.51 *(-20.29)* | 5 / 5 |
| HDX | 1.5 | 83.99 ± 0.00 *(-8.81)* | 48.64 ± 0.00 *(-20.90)* | 22.50 ± 0.00 *(-20.97)* | 5 / 5 |

| $c_{\text{ss,comb}}$ : #parameters $\leq 261650 \wedge$ peak memory usage $\leq 655kB\wedge$ | | | | | |
|---|---|---|---|---|---|
| #FLOPs $\leq 1196M$ *(13956 architectures satisfied)* | | | | | |
| **ProxylessNAS** | **0.5** | **89.56** $_{\pm\,0.44}$ *(-2.81)* | **61.22** $_{\pm\,1.01}$ *(-7.70)* | **30.29** $_{\pm\,0.33}$ *(-11.47)* | 5 / 5 |
| ProxylessNAS | 1.0 | 84.12 $_{\pm\,0.28}$ *(-8.25)* | 49.12 $_{\pm\,1.07}$ *(-19.80)* | 22.64 $_{\pm\,0.31}$ *(-19.13)* | 5 / 5 |
| ProxylessNAS | 1.5 | 84.49 $_{\pm\,0.28}$ *(-7.88)* | 50.56 $_{\pm\,1.07}$ *(-18.36)* | 23.06 $_{\pm\,0.31}$ *(-18.71)* | 5 / 5 |
| FBNet | 0.5 | 92.47 $_{\pm\,0.45}$ | 67.96 $_{\pm\,1.15}$ | 42.33 $_{\pm\,1.48}$ | 0 / 5 |
| FBNet | 0.6 | 92.59 $_{\pm\,0.32}$ | 67.92 $_{\pm\,0.64}$ | 43.09 $_{\pm\,1.12}$ | 0 / 5 |
| FBNet | 0.7 | 92.46 $_{\pm\,0.18}$ | 67.84 $_{\pm\,0.18}$ | 42.41 $_{\pm\,0.48}$ | 0 / 5 |
| TAS | 0.5 | 92.37 $_{\pm\,0.18}$ | 67.40 $_{\pm\,0.51}$ | 41.42 $_{\pm\,0.16}$ | 0 / 5 |
| TAS | 1.0 | 92.50 $_{\pm\,0.00}$ | 67.02 $_{\pm\,0.00}$ | 41.30 $_{\pm\,0.00}$ | 0 / 5 |
| TAS | 1.5 | 91.97 $_{\pm\,0.07}$ | 67.02 $_{\pm\,0.49}$ | 41.08 $_{\pm\,0.29}$ | 0 / 5 |
| **TF-NAS** | **0.5** | **91.36** $_{\pm\,0.38}$ *(-1.01)* | **66.35** $_{\pm\,0.59}$ *(-2.57)* | **38.31** $_{\pm\,0.53}$ *(-3.45)* | 5 / 5 |
| TF-NAS | 1.0 | 90.93 $_{\pm\,0.21}$ *(-1.44)* | 65.41 $_{\pm\,0.37}$ *(-3.51)* | 37.03 $_{\pm\,0.46}$ *(-4.73)* | 5 / 5 |
| TF-NAS | 1.5 | 91.16 $_{\pm\,0.00}$ *(-1.21)* | 65.00 $_{\pm\,0.00}$ *(-3.92)* | 36.53 $_{\pm\,0.00}$ *(-5.23)* | 5 / 5 |
| HDX | 0.5 | 80.63 $_{\pm\,1.88}$ *(-11.74)* | 35.57 $_{\pm\,7.31}$ *(-33.35)* | 16.93 $_{\pm\,3.12}$ *(-24.84)* | 5 / 5 |
| **HDX** | **1.0** | **83.99** $_{\pm\,0.00}$ *(-8.38)* | **48.64** $_{\pm\,0.00}$ *(-20.28)* | **22.50** $_{\pm\,0.00}$ *(-19.27)* | 5 / 5 |
| **HDX** | **1.5** | **83.99** $_{\pm\,0.00}$ *(-8.38)* | **48.64** $_{\pm\,0.00}$ *(-20.28)* | **22.50** $_{\pm\,0.00}$ *(-19.27)* | 5 / 5 |

# E   TRAINING DETAILS

To facilitate reproducibility, we provide additional experimental details in this appendix.

## E.1   NATS-BENCH

We adopt the same data augmentation techniques and training procedure as described in Dong et al. (2021). The search process is performed in two stages.

In the first stage, a supernet is trained on $50\%$ of the CIFAR-10 training set (25,000 images) for 150 epochs, using a batch size of $64$. The operation weights are optimized via Nesterov momentum Stochastic Gradient Descent (SGD) with a momentum of $0.9$ and a weight decay of $5 \times 10^{-4}$. The initial learning rate is set to $0.0025$ and annealed to $0.001$ over 100 epochs using a cosine schedule. Architecture parameters are optimized using Adam with a learning rate of $0.001$, weight decay of $1 \times 10^{-3}$, and $\beta_1 = 0.5$, $\beta_2 = 0.999$. The sampling temperature starts at $10$ and is linearly annealed to $0.1$ over 100 epochs. The rescale factor $R$ is kept constant throughout training (we use $R = 1.2$ for most experiments; an ablation study on the rescale factor is presented in Appendix B).

In the second stage, an architecture is selected based on the lowest cross-entropy loss, evaluated on the remaining $50\%$ of the training set, while satisfying the imposed hardware constraints. The NATS-Bench performance lookup table is then used to retrieve the test accuracy of the sampled architecture on CIFAR-10, CIFAR-100, and ImageNet16-120.

All experiments are repeated five times using fixed random seeds. The search is performed on a single NVIDIA V100 GPU and took approximately 5 hours to complete.

## E.2   CONDITION MONITORING USE CASE

For the condition monitoring use case, a dataset is created from Desenfans et al. (2025), representing readings from a voltage meter, a current sensor and an encoder. (De Ruyter et al., 2025) The complete dataset contains 31,920 time series instances, split into 25,530 train for training and 6,390 for testing. Each instance consists of 256 time steps with 8 features: 3-phase current, 3-phase voltage, motor speed, and rotor angle. The instances are labeled as either *no fault*, *static eccentricity fault*, *dynamic eccentricity fault*, or *mixed eccentricity fault*. A Fast Fourier Transform (FFT) is applied to the 3-phase current and voltage signals, and all 8 features are normalized to the range $[0, 1]$. The search process is again performed in two stages.

In the first stage, a supernet is trained on $80\%$ of the training set (20424 instances) for 150 epochs, using a batch size of $128$. The operation weights are optimized via Nesterov momentum Stochastic Gradient Descent (SGD) with a momentum of $0.9$ and a weight decay of $5 \times 10^{-4}$. The initial learning rate is set to $0.0025$ and annealed to $0.001$ over 100 epochs using a cosine schedule. Architecture parameters are optimized using Adam with a learning rate of $0.001$, weight decay of $1 \times 10^{-3}$, and $\beta_1 = 0.5$, $\beta_2 = 0.999$. The sampling temperature starts at $10$ and is linearly annealed to $0.1$ over 100 epochs. The rescale factor $R$ is kept constant at $1.2$.

In the second stage, an architecture is selected based on the lowest cross-entropy loss, evaluated on the remaining $20\%$ of the training set, while satisfying the imposed hardware constraints. The selected architectures are trained using 5-fold cross-validation, where each fold uses $80\%$ of the training data for training over 250 epochs with a batch size of $16$. Optimization is performed using SGD with a momentum of $0.9$ and a weight decay of $1 \times 10^{-4}$. The initial learning rate is set to $0.0025$ and annealed to $0.0001$ using a cosine scheduler.

All experiments are repeated five times using fixed random seeds. The search is performed on a single NVIDIA P100 GPU and took approximately 7 hours to complete.

## F    Condition Monitoring Search Space

The search space for the condition monitoring use case is defined as a supernet composed of eight consecutive edges. Each edge can select from the following candidate operations:

- A 1D convolution followed by a batch normalization and a ReLU activation
- A 1D depthwise separable convolution, where each convolution is followed by a batch normalization and a ReLU activation
- An identity operation (available only on even-numbered edges)

The classification head consists of a global average pooling layer followed by a fully connected layer.

For each convolution, the number of output channels can be chosen from $\{16, 32, 64, 256, 512\}$. Additionally, a stride of 2 is applied to convolutions on odd-numbered edges. Convolutional filters are shared across operations, as proposed in Wan et al. (2020), to reduce the number of trainable parameters and improve memory and computational efficiency.

## G    CONDITION MONITORING HAND-CRAFTED BASELINES

To ensure coverage of diverse hardware constraints, we sampled six hand-crafted architectures from the proposed search space. These include the smallest (Conv1D-x-Min) and largest (Conv1D-x-Max) models in terms of latency and energy consumption, for both regular and depthwise-separable convolutions (Chollet, 2017). Additionally, Conv1D-Reg and Conv1D-DS follow common design principles where channel width increases as spatial dimensions decrease.

Table 11: **Overview of hand-crafted architectures used for comparison.** Each convolutional block consists of a convolutional operation regular or depthwise separable, a batch normalization, and a ReLU activation.

| Name | Number of blocks | Kernel | Channels | Strides | Convolution type |
|---|---|---|---|---|---|
| Conv1D-Reg | 4 | 3 | [16, 32, 64, 128] | [2, 2, 2, 2] | Regular |
| Conv1D-DS | 4 | 3 | [16, 32, 64, 128] | [2, 2, 2, 2] | Depthwise Separable |
| Conv1D-Reg-Max | 8 | 3 | [512, 512, 512, 512, 512, 512, 512, 512] | [2, 1, 2, 1, 2, 1, 2, 1] | Regular |
| Conv1D-DS-Max | 8 | 3 | [512, 512, 512, 512, 512, 512, 512, 512] | [2, 1, 2, 1, 2, 1, 2, 1] | Depthwise Separable |
| Conv1D-Reg-Min | 4 | 3 | [8, 8, 8, 8] | [2, 2, 2, 2] | Regular |
| Conv1D-DS-Min | 4 | 3 | [8, 8, 8, 8] | [2, 2, 2, 2] | Depthwise Separable |

## H    DISCLOSURE ON THE USE OF GENERATIVE ARTIFICIAL INTELLIGENCE

We used a large language model (LLM) as a writing aid. Ideation, scientific content, experimental design, and analysis were conducted entirely by the authors without any contribution of an LLM.

