# OpenReview forum: "Constraint-guided Hardware-aware NAS through Gradient Modification"
_ICLR.cc/2026/Conference — ICLR 2026 Poster_

### Official Review · Reviewer_Sicz · 2025-10-31

**Soundness:** 4
**Presentation:** 3
**Contribution:** 3
**Rating:** 6
**Confidence:** 5

**Summary:**

This paper proposes to search for neural network architecture in a differentiable manner, guaranteeing hardware constraint compliance.
There are prior works that adopt hardware-related loss to add a hardware related objective.
However, using hardware loss as a penalty doesn't guarantee compliance with the hardware constraint.
Instead of using auxiliary loss for constraint, the paper proposes to modify the gradients of architecture parameters toward a direction for constraint compliance.

The novelty of the paper is computing the direction of the gradient that make the searched network comply with hardware constraint.

For a layer $l$, additional gradients of architecture parameters are achieved in the following manner.
- If all candidate operations comply with the constraint, the additional gradients are zero.
- If only some of them comply, split candidate operations into two sets according to the compliance.
For all combinations made by extracting one from both sets, a unit vector whose values are zero except for the indices of selected operations is initialized.
The absolute values of the indices are the same, while the sign of the value becomes positive if it complies with the constraint, negative if not.
And then, all combinations of the unit vectors are summed, and the result vector is normalized.
- If all candidate operations don't comply with the constraint, sort the operations according to the hardware metric and get all the combinations of two operations among them.
Then, follow the steps in the above.

With the proposed gradient modification, the well-performed network architectures that comply with the given constraint can be searched.

**Strengths:**

- Instead of adopting an auxiliary loss to search models that comply with the given constraint, the paper proposes to modify the gradients of architecture parameters according to the relative relation between arbitrary candidate operations. This idea is strong and valid.

- The performance of searched models is less affected by the hyperparameter that balances the gradient for performance and that for compliance. The hyperparameter doesn't have to be fine-tuned well, and it leads to reduce the search cost, too.

- The paper develops what it contends well, and thus it is easy to follow. Also, the paper shows the effectiveness of the proposal well, with various experiments.

**Weaknesses:**

- The paper addresses many prior works in the NAS field, but it doesn't address hardware constrained NAS works [1, 2], which are directly related to this paper.

- It is hard to understand Figure 1. Are the $G_{\alpha_l}$ and $-\nabla_{\alpha_l} \mathcal{L}(w, a)$ of each graph projecting the same vectors in different dimensions? If so, it is not clearly delivered.


[1] Nayman, Niv, et al. "Hardcore-nas: Hard constrained differentiable neural architecture search." International Conference on Machine Learning. PMLR, 2021.

[2] Hong, Deokki, et al. "Enabling hard constraints in differentiable neural network and accelerator co-exploration." Proceedings of the 59th ACM/IEEE Design Automation Conference. 2022.

**Questions:**

- Are the architecture parameters of all layers updated with the proposed gradient modification at once?
Or, a subset of layers is selected to update their architecture parameters with a single batch?

- The reviewer think that the proposal will be too complicated if there are multiple constraints.
Can the authors execute additional experiments with multiple constraints, and report how much the experiment time increases  and whether the searched networks comply with constraints well?

---

> ### Author Response · Authors · 2025-11-25
> **Response to Reviewer Sicz**
>
> Thank you for the valuable comments. Please find our response to your questions below.
>
> # Sicz-W1
> We will add the suggested references to the related work section and include results for Hong et al. (2022) and Hu et al. (2020).
>
>
> # Sicz-W2
> $G_{\alpha_l}$ and $-\nabla_{\alpha_l}\mathcal{L}(w,\alpha)$ are indeed the same vectors projected onto the architecture space for $\alpha_{l,i}$ and $\alpha_{l,j}$ respectively. However, we identified an error where $G_{\alpha_1}$ was mistakenly used instead of $G_{\alpha_l}$ in some subfigures and in the caption. We will correct this in the final version of the paper.
>
>
> # Sicz-Q1
> The gradient modifications are applied to all edges within a single batch before updating the architecture parameters. We will make this clearer in the text.
>
> # Sicz-Q2
> Table 2 already shows results for combinations of hardware constraints. We provided some additional results for stricter combinations of hardware constraints in the general response section.
>
>
> See general response to all reviewers regarding runtime analysis.

---

> > ### Comment · Reviewer_Sicz · 2025-11-27
> >
> > Thanks to the authors for the kind answers to the reviewer’s questions.
> >
> > The reviewer maintains the stance of acceptance, under the assumption that the authors will update the manuscript as mentioned in the rebuttal.
> >
> > However, the reviewer would like to raise an additional question regarding Q2.
> >
> > The authors answered the question by focusing on the combinations of hardware constraints.
> > What the reviewer wanted to know, however, is how much the search time increases when an additional constraint is applied.
> >
> > Could the authors provide further explanation about how the search time scales with the number of constraints?

---

> > > ### Author Response · Authors · 2025-11-28
> > >
> > > Thank you for the clarification. We let all experiments run for 150 epochs. However, most experiments already found a satisfactory solution earlier. Based on our experimental setup, we observe that the number of constraints has no noticeable impact on the search time.  Note, however, that this may depend on the specific constraints and search space. For example, if two constraints are highly negatively correlated, it may take longer to find a solution that satisfies both. Below we provide the earliest epoch at which the satisfied solution occurred for our experiments:
> > >
> > >
> > >
> > > ### NATS-Bench Topology Search Space
> > >
> > > | Constraints   |  First Occurrence (epoch) |
> > > |---------------|--------------------------|
> > > |$c_{ts, \text{latency}}$: $\text{latency} \leq 5.31ms$ | 92 ± 19 |
> > > |$c_{ts, \text{energy}}$: $\text{energy usage} \leq 23.95mJ$ | 101 ± 32 |
> > > |$c_{ts, \text{combined}}$: $\text{latency} \leq 5.31ms \land \text{energy usage} \leq 23.95mJ$ | 97 ± 13 |
> > >
> > >
> > >
> > > ### NATS-Bench Size Search Space
> > >
> > > | Constraints   |  First Occurrence (epoch) |
> > > |---------------|--------------------------|
> > > |$c_{ss, \text{param}}$: $\text{number parameters} \leq 261650$ | 104 ± 12 |
> > > |$c_{ss, \text{memory}}$: $\text{peak memory usage} \leq 655kB$ | 87 ± 31 |
> > > |$c_{ss, \text{FLOPs}}$: $\text{number FLOPs} \leq 344194M$ | 108 ± 17 |
> > > |$c_{ss, \text{combined}}$: $\text{number parameters} \leq 261650 \land \text{peak memory usage} \leq 655kB \land \text{number FLOPs} \leq 344194M$ | 104 ± 10 |

---

### Official Review · Reviewer_z6HG · 2025-11-01

**Soundness:** 3
**Presentation:** 3
**Contribution:** 2
**Rating:** 6
**Confidence:** 4

**Summary:**

The paper proposes CONNAS, a gradient-based NAS framework that finds the optimal feasible neural network architecture that satisfies a set of hardware constraints (number of parameters, FLOPS, and peak memory usage, etc.). The problem is defined as choosing the optimal subset of operations from a supernet by optimizing both the architecture weight and the operation weight. Unlike existing differentiable methods that include hardware regularization terms in the loss function, CONNAS directly modifies the gradient direction to favor the architecture that satisfies the hardware constraints. For each architecture weight, they find a normalized unit vector by aggregating all directionalities that point from candidates not satisfying the constraint, to the ones that do. Then, the vector is weighted and added to the gradient to push the update away from constraint-unsatisfying candidates, and towards the satisfying ones.
The method is evaluated on two benchmark tasks, NATS-Bench-TS and NATS-Bench-SS, and against gradient-based NAS methods that use hardware-aware regularization terms. The authors show the searched architecture has better predictability whilst better conforming to the given hardware constraints.

**Strengths:**

**[S1]** Simple and modular mechanism for constraint enforcement.
CONNAS steers the architecture update using a geometrically defined direction dir_{C} computed from local candidate substitutions instead of crafting differentiable hardware surrogates and tuning the weights. In addition, CONNAS demonstrates a clean adaptation of CGGD to NAS without performance degradation.

**[S2]** Solid empirical evidence on multiple constraint types and search spaces
On NATS-Bench-TS/SS across CIFAR-10, CIFAR-100, and ImageNet16-120, CONNAS achieves a low relative error compared to the optimal feasible architectures while consistently satisfying constraints. In CSS-MEM, CIFAR-10 shows a 0.14% relative error, and strong results hold for PARAM and FLOPs as well, as shown in Table 2 and Figure 2. Furthermore, the paper reports an ablation on the rescale factor R, demonstrating robustness for moderate R in Appendix B.

**[S3]** Practicality in real scenarios on edge-ML cases.
By presenting experiments on induction-motor fault diagnosis under explicit MCU-level (STM32 family) budgets and within a large 1.94B-architecture 1D-conv search space, the paper shows consistent outperformance over handcrafted baselines under comparable budgets. Providing a practical scenario further strengthens the paper’s motivation for edge-ML applications.

**[S4]** Good writing
The overall presentation of the paper is good and easy to follow. The authors describe their approach in detail with well-defined terms.

**Weaknesses:**

**[W1]** Missing baselines and fairness should be stronger.
The paper does seem to discuss existing solutions. In my search, it appears there are several existing works on hard-constrained NAS, and some of them are very similar to this work, including Hardcore-NAS [1], HDX [2], or TF-NAS [3]. I believe the authors should clearly discuss the difference and additional novelty over these, preferably with experimental comparison.

[1] Nayman, N. et al., "Hardcore-nas: Hard constrained differentiable neural architecture search." ICML, 2021.

[2] Hong, D. et al., “Enabling hard constraints in differentiable neural network and accelerator co-exploration”, DAC, 2022.

[3] Hu, Y. et al., "TF-NAS: Rethinking Three Search Freedoms of Latency-Constrained Differentiable Neural Architecture Search", ECCV, 2020

**[W2]** The experiments lack comparison under varying difficulty of the constraints. The authors mention “The constraints are chosen such that about 50% of the architectures in each search space meet them”, which seems to be a rather easy bar to meet. In my opinion, methods like this would be much more valuable and convincing when the constraints are sufficiently rigorous that handcrafting or sampling methods fail to find a good candidate. Can the authors evaluate their method under different levels of constraint difficulty (e.g., 25%, 10%, or 5% of architectures satisfying the constraints) to better assess its robustness and effectiveness?

**[W3]** Theoretical fallbacks when moving CGGD to architecture parameters should be discussed more.
The convergence statement in Sec3.3 relies on CGGD’s guarantee when R >1 and shortest-path directions yield convergence to the feasibility region. However, when mapping from \alpha to a discrete architecture via Gumbel Softmax annealing and the local replace-one-edge construction of dir_C, the resulting dir_C could violate the assumptions underlying the original convergence guarantee. A formal theorem should strengthen the claim.

**[W4]** Hardware-aware evaluation relies on analytical metrics not device measurements.
For NATS-Bench, hardware metrics are computed analytically, as shown in Appendix A, rather than measured on devices or by learned LUT regressors for a target platform. This condition reduces deployment realism and could introduce bias.

**[W5]** There is a lack of explanation behind how the manually designed baseline architecture was found. When the authors say “best manually designed model”, how was this model obtained? Did it go through a systematic search process? Is it just based on expert intuition? Appendix F only lists the outcome of such handcrafting and does not explain the process. To make the evaluation results more convincing, the authors should detail this process clearly in the paper.

**Questions:**

In general, please refer to weaknesses section. below are some relatively minor ones.
1. It seems figure 2 is not explained anywhere. Does the red dashed line denote 100% accuracy? Or is it a handcrafted or manually designed baseline?
2. Since Fig. 2 excludes infeasible runs, could the authors provide the fraction of runs that reach feasibility within the budget, as well as the epoch at which feasibility is first reached?
3. Could the authors provide the overhead per epoch of computation dir_C relative to vanilla DARTS-like training, especially for large candidate sets per edge? Since there seems to be no analysis for the overhead on computation dir_C, it could make the paper stronger.
4. In algorithm 1, the paper sums and normalizes directions from all violated constraints. What happens when there are conflicting constraints, such as a case where memory and FLOPs push different edges in opposite directions? Is there any regulation regarding multi-constraint interactions?

---

> ### Author Response · Authors · 2025-11-25
> **Response to Reviewer z6HG**
>
> # z6HG-W1
> We will add the suggested references to the related work section. We have also conducted experiments for Hong et al. (2022) and Hu et al. (2020) (see general response).
>
> # z6HG-W2
> See general response regarding additional results on stricter hardware constraints.
>
>
> # z6HG-W3
> While we rely on $R > 1$ and the shortest-path direction (which guarantees convergence in the original CGGD framework), we do not claim that our method guarantees convergence to a feasible solution. We acknowledge that this point may not have been sufficiently clear in the original text and could have caused confusion for readers. The proposed gradient construction for the direction vectors, however, serves as a heuristic that removes the need for full factorization of hardware metrics over architecture parameters. Our experiments demonstrate that this approach effectively guides the search toward feasible solutions in practice. We will clarify this point in the text.
>
> # z6HG-W4
> See general response regarding real hardware evaluation.
>
> # z6HG-W5
> The six hand-crafted architectures are sampled from the proposed search space to cover different hardware constraints. We selected the smallest (Conv1D-X-Min) and largest architectures (Conv1D-X-Max) in terms of latency and energy consumption, both for regular and depthwise-separable convolutions. Additionally, we included two architectures based on common design principles where channels increase as spatial dimensions decrease. By “best manually designed architecture,” we refer to the architecture with the highest accuracy among these hand-crafted architectures (Conv1D-Reg). We will clarify this in the final version.
>
> # z6HG-Q1
> The red-dashed line indicates the the best architecture in the search space with the given hardware constraint. However, this line is indeed redundant, since the best architecture will always have 0% relative error against itself and was not intended to be included in the plot. We will remove this line in the final version of the paper.
>
> # z6HG-Q2
> The number of satisfying runs for each method is summarized in Table 2 under the "satisfied" column, as well as some additional experiments with different scale factors in Table 7 in Appendix D. We also decided to include runs that do not satisfy the constraints in the figure of the final version of the paper, as omitting them might cause confusion for readers.
>
> # z6HG-Q3
> See general response to all reviewers regarding runtime analysis.
>
> # z6HG-Q4
> Currently, all constraints are treated equally regardless of their scale. This is because we normalize each direction vector to have unit norm. However, it is possible to weight the direction vectors based on the relative importance of each constraint or calculate an aggregate score based on how many constraints an edge would violate. We have not explored this in the current work but will consider this as a potential direction for future research.

---

### Official Review · Reviewer_41Ma · 2025-11-02

**Soundness:** 4
**Presentation:** 4
**Contribution:** 2
**Rating:** 4
**Confidence:** 4

**Summary:**

This paper proposes CONNAS, a gradient-based hardware-aware NAS method that enforces resource constraints through constraint-guided gradient modification rather than loss regularization. Instead of adding differentiable hardware penalties, CONNAS projects architecture gradients toward the feasible region, avoiding hyperparameter tuning of penalty weights.

Evaluations on NATS-Bench (CIFAR-10/100, ImageNet16-120) and a real STM32 MCU deployment show that CONNAS consistently meets hardware limits while achieving accuracy close to or better than regularization-based methods (ProxylessNAS, FBNet, TAS). It is robust to hyperparameters and effective under tight constraints.

Overall: An effective approach to enforce hardware constraints directly in differentiable NAS. The idea is empirically validated and practical for real deployment scenarios.

**Strengths:**

1. Clear and well-organized presentation. The paper is written in a clean, logical manner, making both the motivation and method easy to follow.

2. Experimental results on multiple benchmarks and a real hardware deployment convincingly demonstrate the effectiveness and robustness of the proposed CONNAS framework.

3. The idea of modifying gradients directly to satisfy hardware constraints, rather than learning or regularizing them, is unusual, providing a new methodological angle for hardware-aware NAS.

**Weaknesses:**

1. Outdated baselines and limited impact relative to a mature research field. Most compared baselines (ProxylessNAS, FBNet, TAS) are from 2019–2020, while recent NAS frameworks or one-shot supernet approaches are missing. Given that hardware-aware NAS is now a mature area, the lack of modern baselines (e.g., SPOS variants, DiffNAS, or once-for-all models) somewhat limits the significance and contemporary relevance of the work.

2. Directly modifying the gradients, although rare, in principle seems less elegant than learning-based methods. The authors argue that learning-based approaches require tuning the weights of additional penalty terms, but the proposed gradient modification also introduces extra hyperparameters, such as the rescale factor that determines the strength of constraint enforcement. Therefore, to me, the learning-based methods appear more elegant and principled overall.

**Questions:**

See weaknesses.

---

> ### Author Response · Authors · 2025-11-25
> **Response to Reviewer 41Ma**
>
> Thank you for the valuable comments. Please find our response to your questions below.
>
> # 41Ma-W1
> The primary focus of our work is to integrate constrained optimization techniques into gradient-based NAS to directly handle hardware constraints. To evaluate this contribution, we selected well-established hardware-aware NAS methods (such as ProxylessNAS, FBNet, ...) as baselines because we noticed that they are widely used in hardware-aware NAS literature. We acknowledge that one-shot approaches such as Once-for-All and SPOS are relevant baselines, although they also date from the same period (e.g., Cai et al., “Once-for-All: Train One Network and Specialize It for Efficient Deployment,” ICLR 2020). We excluded these methods to maintain a fair comparison with approaches that optimize architecture parameters via gradient descent, similar to our own.
>
> While we agree that hardware-aware NAS is a mature field, we believe that our contribution remain valuable, especially since our method of explicitly modifying gradients to enforce constraints is novel and has not been extensively explored in the literature. As noted in our paper, our method is orthogonal to the choice of gradient-based NAS algorithm. This means that ConNAS could, in principle, be applied to frameworks such as SPOS, where searching through the supernet is performed via gradient descent rather than sampling-based methods like evolutionary search. Exploring this integration would be an interesting direction for future work.
>
> # 41Ma-W2
> The rationale of directly modifying gradients to enforce hardware constraints is motivated by the difficulties of tuning penalty weights in many traditional loss-based approaches. As shown in Appendix B, ConNAS demonstrates strong robustness to the choice of rescale factor, consistently satisfying constraints across different settings. While our method does also introduce a hyperparameter, its sensitivity is far lower compared to penalty weights in loss-based methods (Appendix D). Furthermore, our approach removes the requirement for differentiable hardware metrics or full factorization over architecture parameters.

---

### Official Review · Reviewer_wD6Z · 2025-11-02

**Soundness:** 2
**Presentation:** 2
**Contribution:** 2
**Rating:** 4
**Confidence:** 4

**Summary:**

This paper presents CONNAS, a constraint-guided gradient modification framework for hardware-aware neural architecture search (NAS). Instead of introducing differentiable hardware regularizers or manually tuned weighting factors, CONNAS modifies the gradients with respect to architectural parameters to directly enforce hardware constraints such as FLOPs, parameter count, and memory usage. The approach is evaluated on NATS-Bench across multiple datasets (CIFAR-10, CIFAR-100, and ImageNet16-120).

**Strengths:**

1. Clear motivation and practical focus: The paper identifies a well-defined challenge in hardware-aware NAS, balancing accuracy with strict resource constraints, and proposes a solution that directly targets constraint satisfaction.

2. Readable and well-organized writing: The manuscript is well-structured, and the figures and tables are clear, making the technical content accessible without ambiguity.

3. Comparison with relevant baselines: The evaluation includes established gradient-based NAS methods (e.g., ProxylessNAS, FBNet, TAS) re-implemented under a consistent setup, allowing fair comparisons.

**Weaknesses:**

1. Limited experiments: The experiments are only conducted on NATS-Bench, and only small-scale datasets are evaluated. This makes the generality and scalability of the proposed method unclear.

2. Technical novelty: Based on the method description, the gradient update mechanism is primarily derived from CGGD (Van Baelen & Karsmakers, 2023). More clarification on what is newly introduced in applying this approach to NAS is needed to justify the novelty of the contribution.

3. Lack of real hardware evaluation: The so-called hardware-aware metrics are not measured on actual hardware. FLOPs and the number of parameters are theoretical estimates, and the peak memory usage is computed by summing the input and output feature map sizes for each layer, without considering real deployment environments or memory behaviors on target devices.

4. Typos: “seperable” → should be “separable”

**Questions:**

See Weaknesses

---

> ### Author Response · Authors · 2025-11-25
> **Response to Reviewer wD6Z**
>
> Thank you for the valuable comments. Please find our response to your questions below.
>
> # wD6Z-W1
> While we acknowledge that broader experiments would strengthen the evidence for ConNAS’s generality and scalability, we focused on NATS-Bench and small-scale datasets to ensure a thorough evaluation within our computational and time constraints. Additionally, we incorporated a real-world use case beyond traditional literature vision tasks to demonstrate ConNAS’s practicality and effectiveness outside benchmark settings, which we believe partially mitigates concerns about generality. However, we encourage future work to explore ConNAS on larger-scale datasets and diverse architectures as computational resources permit. Therefore, we will publish the code upon acceptance and will include the link in the final version of the paper.
>
> # wD6Z-W2
> The original CGGD framework modifies gradients to enforce inequality constraints on model outputs during weight optimization. In our work, we adopt the same principle of gradient modification but apply it in the context of hardware-aware NAS. Instead of updating model weights to steer outputs toward constraint satisfaction, we adjust architecture parameters to discourage configurations that violate hardware constraints.
>
> Our per-edge gradient direction construction (see Section 3.3) is novel in that it works as a heuristic, defining a local direction toward the feasibility region without requiring full factorization of hardware metrics over architecture parameters. This means the gradient direction does not need to be computed across the entire architecture space. Furthermore, integrating these gradient manipulations into the NAS optimization loop distinguishes our approach from the original CGGD framework. We will better clarify these distinctions in the text.
>
> # wD6Z-W3
> Please refer to the general response regarding real hardware evaluation.
>
> # wD6Z-W4
> Thank you for pointing out this typo, we have corrected it in the revised version.

---

### Author Response · Authors · 2025-11-25
**General response to all reviewers**

We thank all reviewers for their valuable comments and suggestions. We address the common concerns raised in the comments below.

---

> ### Author Response · Authors · 2025-11-25
> **Lack of Real Hardware Evaluation**
>
> We replaced theoretical hardware proxies (e.g., FLOPs, parameter count) with empirical measurements of latency and energy consumption obtained from actual runs on a NVIDIA Edge GPU Jetson TX2. These metrics were sourced from HW-NAS-Bench [1], which provides real-world hardware profiling for architectures in the same search space under realistic deployment conditions. The results below show the test performance of architectures found by ConNAS under real hardware constraints. The constraints were selected such that 50% of all architectures in the search space satisfy them. The relative error (shown in parentheses) is computed against the best architecture in the search space that meets the corresponding constraint.
>
> | Constraints | CIFAR10 |CIFAR100| ImageNet16-120 | Runs satisfied |
> |-------------|----------|---------|----------------|----------------|
> |$\text{latency} \leq 5.31ms$ | 93.18 ± 0.09 *(-1.08)* | 70.12 ± 0.16 *(-3.04)* | 40.59 ± 1.20 *(-6.01)*| 5/5 |
> |$\text{energy usage} \leq 23.95mJ$ | 92.97 ± 0.54 *(-1.18)* | 69.51 ± 1.48 *(-2.43)* | 40.32 ± 1.56 *(-6.45)*| 5/5 |
> |$\text{latency} \leq 5.31ms \land \text{energy usage} \leq 23.95mJ$ | 93.14 ± 0.05 *(-1.01)* | 70.21 ± 0.17 *(-1.73)* | 41.45 ± 1.01 *(-5.08)*| 5/5 |
>
> We will include these results in the final version of the paper along with the baseline comparisons.
>
> [1] Li, Chaojian, Zhongzhi Yu, Yonggan Fu, et al. “HW-NAS-Bench: Hardware-Aware Neural Architecture Search Benchmark.” ICLR 2021.

---

> ### Author Response · Authors · 2025-11-25
> **Additional Results on Stricter Hardware Constraints**
>
> We ran additional experiments for both the topology and size search spaces under stricter hardware constraints. The results below show the test performance of architectures found by ConNAS when applying constraints that approximately 25% ($c_{25}$) and 10% ($c_{10}$) of all architectures in the respective NATS-Bench search spaces satisfy.
>
> We will include these results in an additional appendix.
>
> ### NATS-Bench Topology Search Space
>
> | Constraints | CIFAR10 |CIFAR100| ImageNet16-120 | Runs satisfied |
> |-------------|----------|---------|----------------|----------------|
> | $c_{25, latency}$: $\text{latency} \leq 4.14 ms$ | 92.08 ± 0.07 *(-1.97)* |67.09 ± 0.20 *(-4.43)* |39.09 ± 0.72 *(-5.88)* |5/5
> | $c_{10, latency}$: $\text{latency} \leq 3.03 ms$ | 92.05 ± 0.07 *(-0.73)* |67.01 ± 0.20 *(-1.35)* |38.83 ± 0.72 *(-1.61)* |5/5
> | $c_{25, energy}$: $\text{energy usage} \leq 18.6mJ$ | 92.06 ± 0.06 *(-1.98)* |67.09 ± 0.20 *(-4.43)* |39.04 ± 0.68 *(-5.93)* |5/5
> | $c_{10, energy}$: $\text{energy usage} \leq 13.3mJ$ | 92.05 ± 0.07 *(-0.73)* |67.01 ± 0.20 *(-1.35)* |38.83 ± 0.72 *(-1.61)* |5/5
> | $c_{25, combined}$: $\text{latency} \leq 4.14 ms \land \text{energy usage} \leq 18.6mJ$ | 92.10 ± 0.06 *(-1.95)* |67.16 ± 0.16 *(-4.36)* |39.35 ± 0.59 *(-5.61)* |5/5
> | $c_{10, combined}$: $\text{latency} \leq 3.03 ms \land \text{energy usage} \leq 13.3mJ$ | 92.08 ± 0.06 *(-0.70)* |67.20 ± 0.21 *(-1.16)* |39.10 ± 0.70 *(-1.33)* |5/5
>
> ### NATS-Bench Size Search Space
>
> | Constraints | CIFAR10 |CIFAR100| ImageNet16-120 | Runs satisfied |
> |-------------|----------|---------|----------------|----------------|
> |$c_{25, \text{param}}$: $\text{number parameters} \leq 187210$ | 91.33 ± 0.18 *(-0.50)* | 64.03 ± 0.66 *(-2.55)* | 38.25 ± 0.29 *(-1.01)*| 5/5 |
> |$c_{10, \text{param}}$: $\text{number parameters} \leq 129098$ | 90.02 ± 0.61 *(-1.08)* | 60.85 ± 1.97 *(-3.35)* | 33.89 ± 0.48 *(-2.91)*| 5/5 |
> |$c_{25, \text{memory}}$: $\text{peak memory usage} \leq 393kB$ | 92.91 ± 0.08 *(-0.21)* | 69.28 ± 0.24 *(-1.20)* | 43.65 ± 0.10 *(-1.05)*| 5/5 |
> |$c_{10, \text{memory}}$: $\text{peak memory usage} \leq 229kB$ | 92.18 ± 0.19 *(-0.62)* | 68.63 ± 0.32 *(-0.67)* | 40.61 ± 0.78 *(-1.49)*| 5/5 |
> |$c_{25, \text{FLOPs}}$: $\text{number FLOPs} \leq 147235M$ | 91.58 ± 0.12 *(-0.70)* | 64.94 ± 0.60 *(-3.98)* | 39.53 ± 0.97 *(-1.77)*| 5/5 |
> |$c_{10, \text{FLOPs}}$: $\text{number FLOPs} \leq 67063M$ | 90.83 ± 0.33 *(-0.56)* | 63.43 ± 0.65 *(-3.15)* | 35.97 ± 1.00 *(-3.03)*| 5/5 |
> |$c_{25, \text{combined}}$: $\text{number parameters} \leq 187210 \land \text{peak memory usage} \leq 393kB \land \text{number FLOPs} \leq 147235M$ | 91.29 ± 0.15 *(-0.37)* | 64.88 ± 0.54 *(-1.70)* | 38.03 ± 0.09 *(-1.24)*| 5/5 |
> |$c_{10, \text{combined}}$: $\text{number parameters} \leq 129098 \land \text{peak memory usage} \leq 229kB \land \text{number FLOPs} \leq 67063M$ | 89.52 ± 0.10 *(-0.91)* | 61.79 ± 0.47 *(-1.29)* | 32.23 ± 1.54 *(-2.07)*| 5/5 |

---

> ### Author Response · Authors · 2025-11-25
> **Results on Additional Baseline Methods**
>
> We conducted additional experiments comparing ConNAS with two recent hardware-aware NAS methods suggested by some reviewers:
> - **HDX**:  Hong, D. et al., “Enabling hard constraints in differentiable neural network and accelerator co-exploration”, DAC, 2022.
> - **TF-NAS**: Hu, Y. et al., "TF-NAS: Rethinking Three Search Freedoms of Latency-Constrained Differentiable Neural Architecture Search", ECCV, 2020
>
> We will add the results to the discussion in the final version of the paper.

---

> ### Author Response · Authors · 2025-11-25
> **Comparison Runtime Analysis**
>
> We evaluated the runtime of ConNAS under three constraint settings (one, two, and three constraints) and compared it against baseline methods. The constraints were defined as follows:
>
> - $c_1: \text{number parameters} \leq 261650$
> - $c_2:\text{number parameters} \leq 261650 \land \text{peak memory usage} \leq 655kB$
> - $c_3:\text{number parameters} \leq 261650 \land \text{peak memory usage} \leq 655kB \land \text{number FLOPs} \leq 344194M$.
>
> We report the mean and standard deviation over 100 epochs of architecture parameter optimization for a single batch on the NATS-Bench size search space. All measurements were performed on an NVIDIA P100 GPU.
>
> | Method                | $c_1$ (ms)       | $c_2$ (ms)       | $c_3$ (ms)       |
> |-----------------------|------------------|------------------|------------------|
> | ConNAS                | 35.2 ± 6.06      | 35.9 ± 7.16      | 35.6 ± 7.30      |
> | ProxylessNAS          | 18.2 ± 2.19      | 20.0 ± 0.49      | 22.2 ± 2.96      |
> | FBNet                 | 17.9 ± 1.97      | 20.0 ± 3.61      | 22.6 ± 0.94      |
> | TAS                   | 18.4 ± 3.12      | 20.9 ± 0.71      | 23.0 ± 2.65      |
> | TF-NAS                | 18.1 ± 0.86      | 21.4 ± 2.32      | 22.6 ± 2.53      |
> | HDX                   | 36.4 ± 1.64      | 41.0 ± 5.54      | 45.0 ± 5.52      |
>
> ConNAS introduces an overhead due to the gradient modification step compared to loss-based methods. Crucially, this overhead does not increase with the number of constraints. In contrast, loss-based approaches experience higher runtimes as each additional constraint adds a term to the loss function that must be computed during backpropagation. HDX shows the highest runtime because it combines global loss computation with gradient modifications for individual hardware constraints.

---

### Author Response · Authors · 2025-12-02
**Revised Manuscript**

We have updated the manuscript to address the concerns raised by the reviewers. The major revisions are summarized below:

- We replaced theoretical hardware proxies with empirical measurements of latency and energy consumption from HW-NAS-Bench [1] on the NATS-Bench topology search task. These results are presented in Table 2.
- We added experiments for ConNAS and baseline methods under stricter hardware constraints in both the NATS-Bench topology and size search spaces. These results are included in Appendix C and further demonstrate ConNAS’s ability to consistently find valid architectures with a performance close to the best feasible solution under the given constraints.
- Following reviewer suggestions, we included two additional hardware-aware NAS methods into our comparisons:
	- HDX [2]
	- TF-NAS [3]
- We revised several sections to improve clarity, including:
	- The distinction between our approach and the original CGGD framework (Section 3.3)
	- The definition of hand-crafted architectures for the practical use case (Appendix G)
	- We updated Figure 1 and Figure 2 along with their captions.
- We updated the results for FBNet to reflect its official implementation and hyperparameters.

While we regret that an interactive discussion with the reviewers was not possible due to the recent OpenReview incident, we hope these revisions adequately address all concerns. We would like to take the opportunity to thank the reviewers again for their valuable feedback and suggestions.

[1] Li, Chaojian, Zhongzhi Yu, Yonggan Fu, et al. “HW-NAS-Bench: Hardware-Aware Neural Architecture Search Benchmark.” ICLR 2021.

[2] Hong, D. et al., “Enabling hard constraints in differentiable neural network and accelerator co-exploration”, DAC, 2022.

[3] Hu, Y. et al., "TF-NAS: Rethinking Three Search Freedoms of Latency-Constrained Differentiable Neural Architecture Search", ECCV, 2020

---

### Meta-Review · Area_Chair_6tXr · 2026-01-06

**Summary:**

This paper proposed a novel gradient-based NAS method. It adopt CGGD framework and enforce the hardware constraints directly into the gradient computation for the optimization task. A careful design of gradient direction for the hardware constraints are also proposed. Experiments on NATS-Bench across multiple datasets demonstrate the effectiveness of the proposed method. Reviewers acknowledges that the idea of this paper is novel and experimental results are solid. The paper is also clear and well organized. Major concerns from the reviewers are: 1. The hardware metrics used are not real but theoretically estimated; 2. Lack of comparisons with recently baselines; 3. Lack of comparisons under varying difficulty of constraints. The authors addresses these concerns in the rebuttal making it a qualified publication.

**Reviewer Concerns:**

Major concerns mentioned above are well addressed in the response from the authors:
1. They used real hardware metrics like latency in ms instead of theoretically estimated ones.
2. They include comparisons with literatures mentioned by the reviewers.
3. They include a table with varying constraints.

One concern that is still outstanding is the W3 from reviewer z6HG. The reviewer asks about the "Theoretical fallbacks" when using CGGD, the authors claim that the proposed method has no theoretical guarantee of convergence to a feasible solution but in practice, the  result architecture usually satisfy the constraint. This is an acceptable explanation but the concern may not be fully addressed.

**Reviewer Scores:**

This paper get scores of 4, 4, 6, 6. As the authors addressed most of the concerns, I feel the reviewers may raise their scores. I lean towards accepting this paper.

---

### Decision · Program_Chairs · 2026-01-26

Accept (Poster)